# Approach-avoidance reinforcement learning as a translational and computational model of anxiety-related avoidance

**Yumeya Yamamori[1], Oliver J Robinson[1,2†], Jonathan P Roiser[1*†]**

[1]Institute of Cognitive Neuroscience, University College London, London, United Kingdom; [2]Research Department of Clinical, Educational and Health Psychology, University College London, London, United Kingdom

**Abstract** Although avoidance is a prevalent feature of anxiety-related psychopathology, differences in the measurement of avoidance between humans and non-human animals hinder our progress in its theoretical understanding and treatment. To address this, we developed a novel translational measure of anxiety-related avoidance in the form of an approach-avoidance reinforcement learning task, by adapting a paradigm from the non-human animal literature to study the same cognitive processes in human participants. We used computational modelling to probe the putative cognitive mechanisms underlying approach-avoidance behaviour in this task and investigated how they relate to subjective task-induced anxiety. In a large online study (n = 372), participants who experienced greater task-induced anxiety avoided choices associated with punishment, even when this resulted in lower overall reward. Computational modelling revealed that this effect was explained by greater individual sensitivities to punishment relative to rewards. We replicated these findings in an independent sample (n = 627) and we also found fair-to-excellent reliability of measures of task performance in a sub-sample retested 1 week later (n = 57). Our findings demonstrate the potential of approach-avoidance reinforcement learning tasks as translational and computational models of anxiety-related avoidance. Future studies should assess the predictive validity of this approach in clinical samples and experimental manipulations of anxiety.

**\*For correspondence:**
j.roiser@ucl.ac.uk

[†]These authors contributed equally to this work

## eLife assessment

This is a **valuable** paper demonstrating the validity of a novel task that could advance the field of reinforcement learning to better incorporate threat processing in approach-avoidance-conflict. A **compelling** methodology includes the use of online samples and computational modelling, psycho-metrics, discovery/replication and pre-registration. This work provides a foundation for future work, which is required to establish this task as relevant to psychopathology and treatment.

## Introduction

Avoiding harm or potential threats is essential for our well-being and survival, but it can some-times lead to negative consequences in and of itself. This occurs when avoidance is costly, that is, when the act to avoid harm or threat requires the sacrifice of something positive. When offered the opportunity to attend a job interview, for example, the reward of landing a new job must be weighed against the risk of experiencing rejection and/or embarrassment due to fluffing the

interview. On balance, declining a single interview would likely have little impact on one's life, but routinely avoiding job interviews would jeopardise one's long-term professional opportunities. More broadly, consistent avoidance in similar situations, which are referred to as involving *approach-avoidance conflict*, can have an increasingly negative impact as the sum of forgone potential reward accumulates, ultimately leading to missed opportunities in life. Such excessive avoidance is a hallmark symptom of pathological anxiety (*Barlow, 2002*) and avoidance biases during approach-avoidance conflict are thought to drive the development and maintenance of anxiety-related (*Hayes, 1976*; *Stein and Paulus, 2009*; *Aupperle and Paulus, 2010*; *Mkrtchian et al., 2017*; *Struijs et al., 2018*) and other psychiatric disorders (*Aldao et al., 2010*; *Kakoschke et al., 2019*).

How is behaviour during approach-avoidance conflict measured quantitatively? In non-human animals, this can be achieved through behavioural paradigms that induce a conflict between natural approach and avoidance drives. For example, in the 'Geller-Seifter conflict test' (*Geller et al., 1962*), rodents decide whether or not to press a lever that is associated with the delivery of both food pellets and shocks, thus inducing a conflict. Anxiolytic drugs typically increase lever presses during the test (*Treit et al., 2010*), supporting its validity as a model of anxiety-related avoidance. Consequently, this test along with others inducing approach-avoidance conflict are often used as non-human animal tests of anxiety (*Griebel and Holmes, 2013*). In humans, on the other hand, approach-avoidance conflict has historically been measured using questionnaires such as the Behavioural Inhibition/Activation Scale (*Carver and White, 1994*), or cognitive tasks that rely on motor/response time biases, for example by using joysticks to approach/move towards positive stimuli and avoid/move away from negative stimuli (*Guitart-Masip et al., 2012*; *Phaf et al., 2014*; *Kirlic et al., 2017*; *Mkrtchian et al., 2017*).

Translational approaches, specifically when equivalent tasks are used to measure the same cognitive construct in humans and non-human animals, benefit the study of avoidance and its relevance to mental ill-health for two important reasons (*Bach, 2022*; *Pike et al., 2021*). First, precise causal manipulations of neural circuitry such as chemo/optogenetics are only feasible in non-human animals, whereas only humans can verbalise their subjective experiences – it is only by using translational measures that we can integrate data and theory across species to achieve a comprehensive mechanistic understanding. Second, translational paradigms can make the testing of novel anxiolytic drugs more efficient and cost-effective, since those that fail to reduce anxiety-related avoidance in humans could be identified earlier during the development pipeline.

Since the inception of non-human animal models of psychiatric disorders, researchers have recognised that it is infeasible to fully recapitulate a disorder like generalised anxiety disorder in non-human animals due to the complexity of human cognition and subjective experience. Instead, the main strategy has been to reproduce the physiological and behavioural responses elicited under certain scenarios across species, such as during fear conditioning (*Craske et al., 2006*; *Milad and Quirk, 2012*). Computational modelling of behaviour (*Kriegeskorte and Douglas, 2018*) allows us to take a step further and probe the *mechanisms* underlying such physiological and behavioural responses, using a mathematically principled approach. A fundamental premise of this approach is that the brain acts as an information-processing organ that performs computations responsible for observable behaviours, including approach and avoidance (for a recent review on the application of computational methods to approach-avoidance conflict, see *Letkiewicz et al., 2023*). With respect to translational models, it follows that a valid translational measure of some behaviour not only matches the observable responses across species, but also their underlying computational mechanisms. This criterion, which has been termed *computational validity*, has been proposed as the primary basis for evaluating the validity of translational measures (*Redish et al., 2022*).

To exploit the advantages of translational and computational approaches, there has been a recent surge in efforts to create relevant approach-avoidance conflict tasks in humans, based on non-human animal models (referred to as back-translation; *Kirlic et al., 2017*). One stream of research has focused on exploration-based paradigms, which involve measuring how participants spend their time in environments that include potentially anxiogenic regions. Rodents typically avoid such regions, which can take the form of tall, exposed platforms as in the elevated plus maze (*Pellow et al., 1985*), or large open spaces as in the open-field test (*Hall, 1934*). Similar behaviours have been observed in humans when virtual reality was used to translate the elevated

plus maze (*Biedermann et al., 2017*), and when participants were free to walk around a field or around town (*Walz et al., 2016*). These translational tasks have excellent face validity in that they involve almost identical behaviour compared to their non-human analogues, and their predictive validity is supported by findings that individual differences in self-reported anxiety are associated with greater avoidance of the exposed/open regions of space (*Walz et al., 2016*; *Biedermann et al., 2017*). But what of their computational validity? The computations underlying free movement are difficult to model as the action space is large and complex (consisting of two-dimensional directions, velocity, etc.), which makes it difficult to understand the precise cognitive mechanisms underlying avoidance in these tasks.

Other studies have designed tasks that emulate operant conflict paradigms in which non-human participants learn to associate certain actions with both reward and punishment, such as the Geller-Seifter and Vogel conflict tests (*Geller et al., 1962*; *Vogel et al., 1971*). These studies used decision-making tasks in which participants choose between offers consisting of both reward and punishment outcomes (*Aupperle et al., 2011*; *Sierra-Mercado et al., 2015*). The strengths of such decision tasks lie in the simplicity of their action space (accept/reject an offer; choose offer A or B), which facilitates the computational modelling of decision-making, through for example value-based logistic regression models (*Ironside et al., 2020*) or drift-diffusion modelling (*Pedersen et al., 2021*). However, these tasks have less face validity compared to exploration-based tasks, as explicit offers ('Will you accept £1 to incur an electric shock?') are incomprehensible to the majority of non-human animals (although non-human primates can be trained to do so, see: *Sierra-Mercado et al., 2015*; *Ironside et al., 2020*). Instead, non-human participants such as rodents typically have to learn the possible outcomes given their actions, for example that a lever press is associated with both food pellets and electric shocks. Critically, this means that humans performing offer-based decision-making are doing something *computationally dissimilar* to non-human participants during operant conflict tasks, and the criterion of computational validity is unmet. Moreover, evidence from economic decision-making suggests that explicit offers of probabilistic outcomes can impact decision-making differently compared to when probabilistic contingencies need to be learned from experience (referred to as the 'description-experience gap'; *Hertwig and Erev, 2009*); this finding raises potential concerns regarding the use of offer-based tasks in humans as approximations of non-human tasks that do not involve explicit offers.

Therefore, in the current study, we developed a novel translational task that was specifically designed to mimic the computational processes involved in non-human operant conflict tasks, and was also amenable to computational modelling of behaviour. During the task, participants chose between options that were asymmetrically associated with rewards and punishments, such that the more rewarding options were more likely to result in punishment. However, in contrast to previous translations of these tests using purely offer-based designs, participants had to learn the likelihoods of receiving rewards and/or punishments from experience, thus ensuring computational validity.

Preserving the element of learning in our translational task did not necessarily risk confounding the study of approach-avoidance behaviour. Rather, a critical advantage of our approach was that we could model the cognitive processes underlying approach-avoidance behaviour using classical reinforcement learning algorithms (*Sutton and Barto, 2018*), and then use these algorithms to disentangle the relative contributions of learning and value-based decision-making to anxiety-related avoidance. Specifically, using the reinforcement learning framework; we accounted for individual differences in learning by estimating learning rates, which govern the degree to which recently observed outcomes affect subsequent actions; and we explained individual differences in value-based decision-making using outcome sensitivity parameters, which model the subjective values of rewards and punishments for each participant.

We therefore aimed to develop a new translational measure of approach-avoidance conflict. Based on previous findings that anxiety predicts avoidance biases under conflict (*Walz et al., 2016*; *Biedermann et al., 2017*), we predicted that anxiety would increase avoidance responses (and conversely decrease approach responses) and that this would be reflected in changes in computational parameters. Having found evidence for this in a large initial online study, we retested the prediction that anxiety-related avoidance is explained by greater sensitivity to punishments relative to rewards in a pre-registered independent replication sample.

## Results

### The approach-avoidance reinforcement learning task

We developed a novel approach-avoidance conflict task, based on the 'restless bandit' design (*Daw et al., 2006*), in which the participants' goal was to accrue rewards whilst avoiding punishments (*Figure 1a*). On each of 200 trials, participants chose one of two options, depicted as distinct visual stimuli, to obtain certain outcomes. There were four possible outcomes: a monetary reward (represented by a coin); an aversive sound consisting of a combination of a female scream and a high-pitched sound; both the reward and aversive sound; or no reward and no sound (*Figure 1b*). The options were asymmetric in their associated outcomes, such that one option (which we refer to as the 'conflict' option) was associated with both the reward and aversive sound, whereas the other (the 'safe' option) was only associated with the reward (*Figure 1c*). In other words, choosing the conflict option could result in any of the four possible outcomes on any given trial, but the safe option could only lead to either the reward or no reward, and never the aversive sounds.

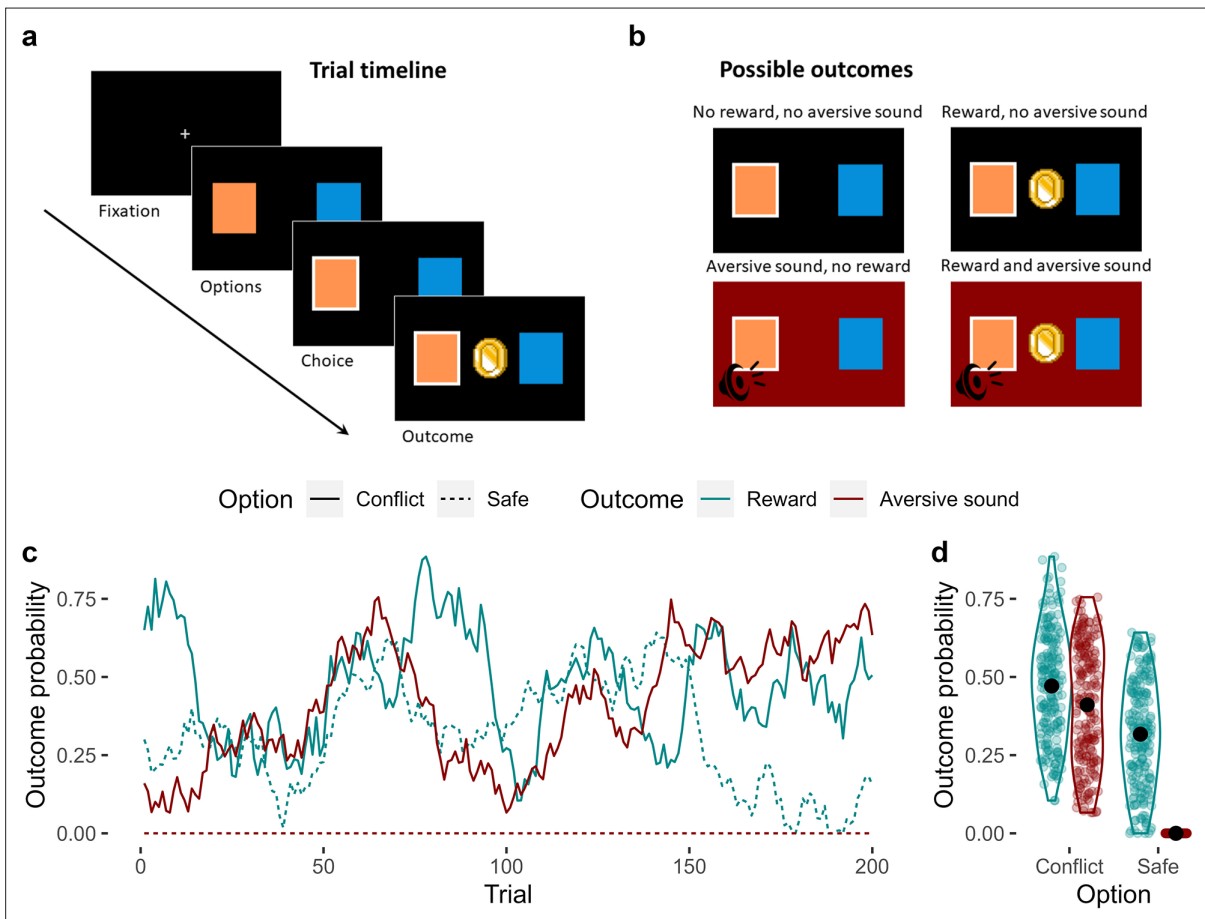

**Figure 1.** The approach-avoidance reinforcement learning task. (**a**) Trial timeline. A fixation cross initiates the trial. Participants are presented with two options for up to 2 s, from which they choose one. The outcome is then presented for 1 s. (**b**) Possible outcomes. There were four possible outcomes: (1) no reward and no aversive sound; (2) a reward and no aversive sound; (3) an aversive sound and no reward; or (4) both the reward and the aversive sound. (**c**) Probabilities of observing each outcome given the choice of option. Unbeknownst to the participant, one of the options (which we refer to as the 'conflict' option – solid lines) was generally more rewarding compared to the other option (the 'safe' option – dashed line) across trials. However, the conflict option was also the only option of the two that was associated with a probability of producing an aversive sound (the probability that the safe option produced the aversive sound was 0 across all trials). The probabilities of observing each outcome given the choice of option fluctuated randomly and independently across trials. The correlations between these dynamic probabilities were negligible (mean Pearson's *r* = 0.06). (**d**) Distribution of outcome probabilities by option and outcome. On average, the conflict option was more likely to produce a reward than the safe option. The conflict option had a variable probability of producing the aversive sound across trials, but this probability was always 0 for the safe option. Black points represent the mean probability.

The probability of observing the reward and the aversive sound at each option fluctuated randomly and independently over trials, except the probability of observing the aversive sound at the safe option, which was set to 0 across all trials (*Figure 1c*). On average, the safe option was less likely to produce the reward compared to the conflict option across the task (*Figure 1d*). Therefore, we induced an approach-avoidance conflict between the options as the conflict option was generally more rewarding but also more likely to result in punishment. Participants were not informed about this asymmetric feature of the task, but they were told that the probabilities of observing the reward or the sound were independent and that they could change across trials.

We initially investigated behaviour in the task in a discovery sample (n = 369) after which the key effects-of-interest were retested in a pre-registered study (https://osf.io/8dm95) with an independent replication sample (n = 629). The discovery sample had a significantly greater proportion of female participants than the replication sample (59% vs 52%, $\chi^2$ = 4.64, *p* = 0.031). The average age was also significantly different across samples (discovery sample mean = 37.7, SD = 10.3, replication sample mean = 34.3, SD = 10.4; $t_{785.5}$ = 5.06, *p* < 0.001). The differences in self-reported psychiatric symptoms across samples did not reach significance (*p* > 0.086). Since our behavioural findings were broadly comparable across samples, we primarily report findings from the discovery sample and discuss deviations from these findings in the replication sample. A detailed comparison of statistical findings across samples is reported in Appendix 1.

## Choices reflect both approach and avoidance

First, we verified that participants demonstrated both approach and avoidance responses during the task through a hierarchical logistic regression model, in which we used the latent outcome probabilities to predict trial-by-trial choice. Overall, participants learned to optimise their choices to accrue rewards and avoid the aversive sounds. As expected, across trials participants chose (i.e. approached) the option with relatively higher reward probability ($\beta$ = 0.98 ± 0.03, *p* < 0.001; *Figure 2a and c*). At the same time, they avoided the conflict option when it was more likely to produce the punishment ($\beta$ = –0.33 ± 0.04, *p* < 0.001; *Figure 2a and d*). These effects were also clearly evident in our replication sample (effect of relative reward probability: $\beta$ = 0.95 ± 0.02, *p* < 0.001; effect of punishment probability: $\beta$ = –0.52 ± 0.03, *p* < 0.001; *Appendix 1—table 1*). While sex was significantly associated with choice in the hierarchical logistic regression in the discovery sample ($\beta$ = 0.16 ± 0.07, *p* = 0.028) with males being more likely to choose the conflict option, this pattern was not evident in the replication sample ($\beta$ = 0.08 ± 0.06, *p* = 0.173), and age was not significantly associated with choice in either sample (*p* > 0.2). Across individuals, there was considerable variability in overall choice proportions (discovery sample: mean = 0.52, SD = 0.14, min/max = [0.03, 0.96]; replication sample: mean = 0.52, SD = 0.15, min/max = [0.01, 0.99]).

## Task-induced anxiety is associated with greater avoidance

Immediately after the task, participants rated their task-induced anxiety in response to the question, 'How anxious did you feel during the task?', from 'Not at all' to 'Extremely' (scored from 0 to 50). On average, participants reported a moderate level of task-induced anxiety (mean rating of 21, SD = 14; *Figure 2b*).

Task-induced anxiety was significantly associated with multiple aspects of task performance. At the most basic level, task-induced anxiety was correlated with the proportion of conflict/safe option choices made by each participant across all trials, with greater anxiety corresponding to a preference for the safe option over the conflict option (permutation-based non-parametric correlation test: Kendall's $\tau$ = –0.074, *p* = 0.033), at the cost of receiving fewer rewards on average.

To investigate this association further, we included task-induced anxiety into the hierarchical logistic regression model of trial-by-trial choice. Here, task-induced anxiety significantly interacted with punishment probability ($\beta$ = –0.10 ± 0.04, *p* = 0.022; *Figure 2a*), such that more anxious individuals were proportionately less willing to choose the conflict option during times it was likely to produce the punishment, relative to less-anxious individuals (*Figure 2e*). This explained how task-induced anxiety influenced participants' choices, as the main effect of anxiety was not significant in the model, although it was when the interaction effects were excluded. The analogous interaction between task-induced anxiety and relative reward probability was non-significant ($\beta$ = –0.06 ± 0.03, *p* = 0.074, *Figure 2a*).

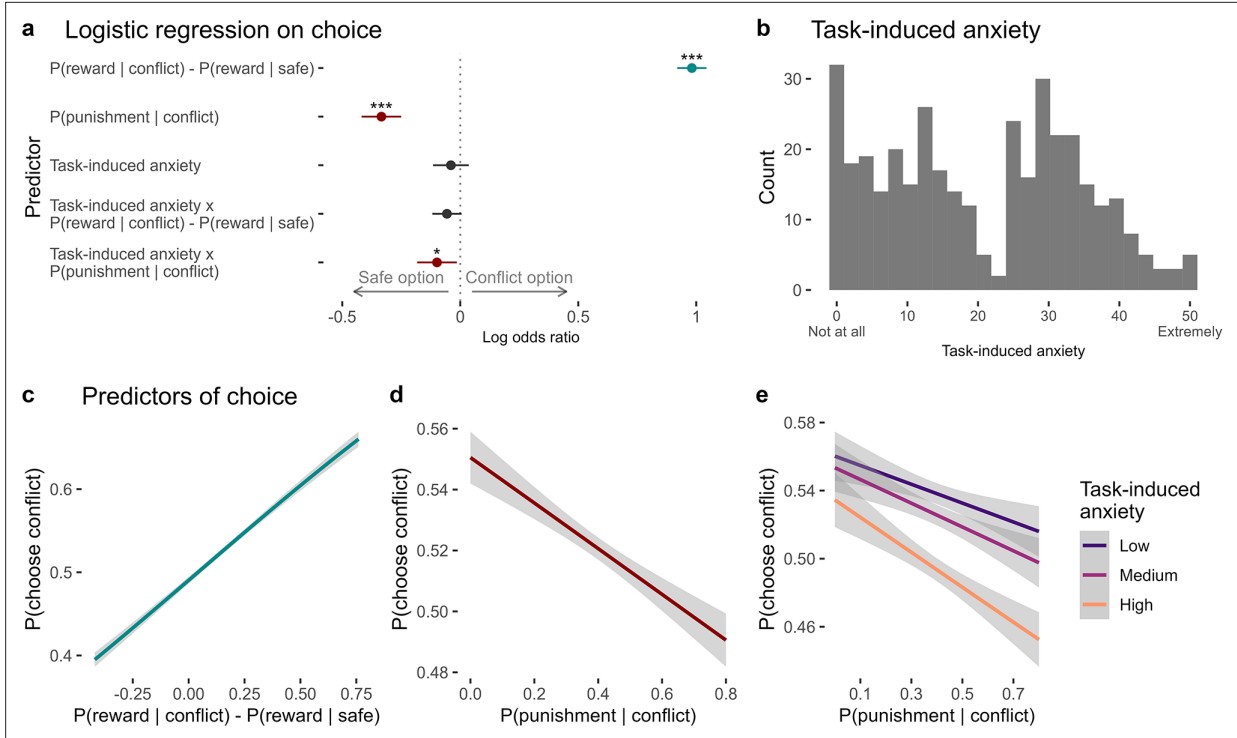

**Figure 2.** Predictors of choice in the approach-avoidance reinforcement learning task. (**a**) Coefficients from the mixed-effects logistic regression of trial-by-trial choices in the task (n = 369). On any given trial, participants chose the option that was more likely to produce a reward. They also avoided choosing the conflict option when it was more likely to produce the punishment. Task-induced anxiety significantly interacted with punishment probability. Significance levels are shown according to the following: $p < 0.05$ – *; $p < 0.01$ – **; $p < 0.001$ – ***. Error bars represent confidence intervals. (**b**) Subjective ratings of task-induced anxiety, given on a scale from 'Not at all' (0) to 'Extremely' (50). (**c**) On each trial, participants were likely to choose the option with greater probability of producing the reward. (**d**) Participants tended to avoid the conflict option when it was likely to produce a punishment. (**e**) Compared to individuals reporting lower anxiety during the task, individuals experiencing greater anxiety showed greater avoidance of the conflict option, especially when it was more likely to produce the punishment. Note. Figures c–e show logistic curves fitted to the raw data using the 'glm' function in R. For visualisation purposes, we categorised continuous task-induced anxiety into tertiles. We show linear curves here since these effects were estimated as linear effects in the logistic regression models, however the raw data showed non-linear trends – see *Appendix 11—figure 1*.

These associations with task-induced anxiety were again clearly evident in the replication sample (correlation between task-induced anxiety and proportion of conflict/safe option choices: $\tau = -0.075$, $p = 0.005$; interaction between task-induced anxiety and punishment probability in the hierarchical model: $\beta = -0.23 \pm 0.03$, $p < 0.001$; *Appendix 1—table 1*).

## A reinforcement learning model explains individual choices

Next, we investigated the putative cognitive mechanisms driving behaviour on the task by fitting standard reinforcement learning models to trial-by-trial choices. The winning model in both the discovery and replication samples included reward- and punishment-specific learning rates, and reward- and punishment-specific sensitivities (i.e. four parameters in total; *Figure 3a and b*). Briefly, it models participants as learning four values, each corresponding to the probabilities of observing specific outcomes (reward or punishment) associated with each option (conflict or safe). At every trial, the probability estimates for the chosen option are updated after observing the outcome(s) according to the Rescorla-Wagner update rule. Individual differences in the rate of updating are captured by the learning rate parameters ($\alpha^r$, $\alpha^p$), whereas differences in the extent to which each outcome impacts choice are captured by the sensitivity parameters ($\beta^r$, $\beta^p$).

To assess how well this model captured our observed data, we simulated data from the model given each participant's parameter values, choices, and observed outcomes. These synthetic choice data closely resembled the actual choices made by the participants (*Figure 3c*). Parameter recovery from the winning model was excellent, with Pearson's $r$ values between data-generating and recovered parameters ranging from 0.77 to 0.86 (*Appendix 4—figure 1*).

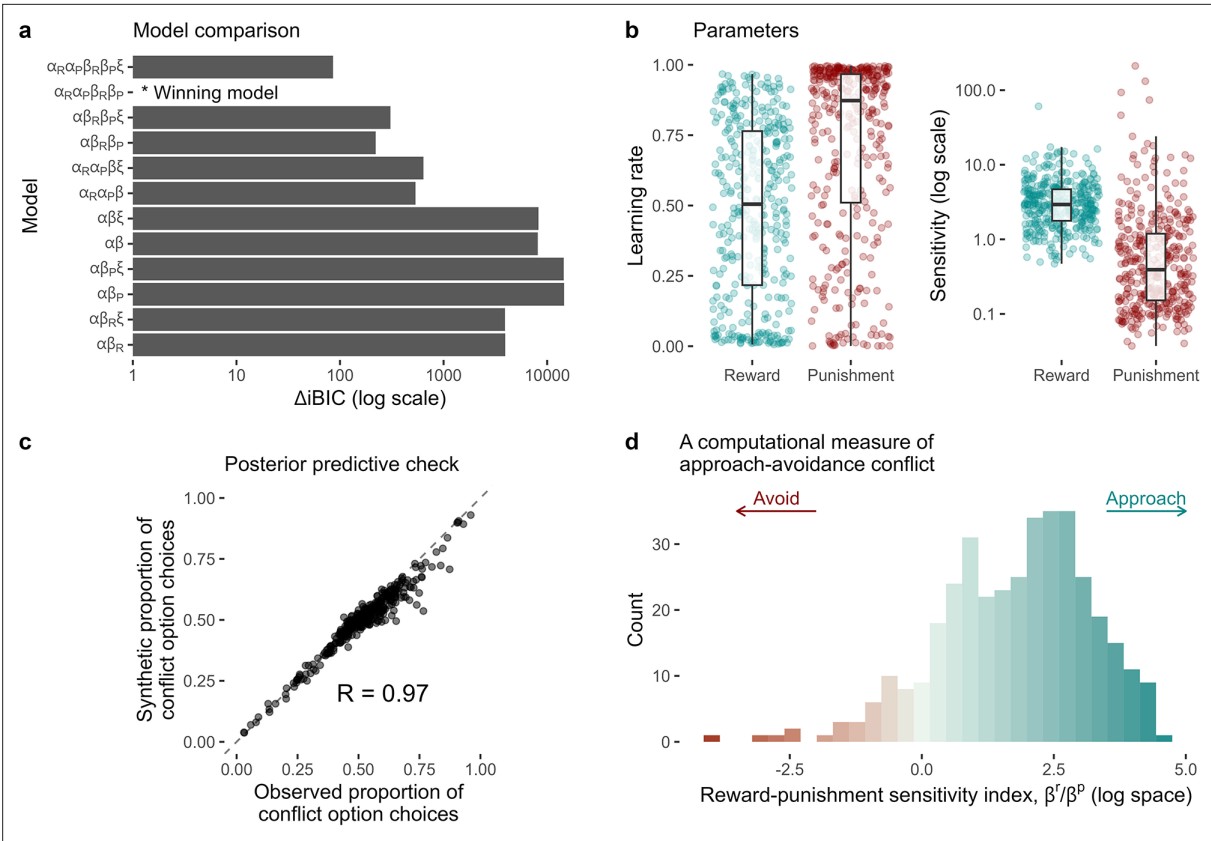

**Figure 3.** Computational modelling of approach-avoidance reinforcement learning. (**a**) Model comparison results (n = 369). The difference in integrated Bayesian information criterion scores from each model relative to the winning model is indicated on the x-axis. The winning model included specific learning rates for reward ($\alpha^R$) and punishment learning ($\alpha^p$), and specific outcome sensitivity parameters for reward ($\beta^R$) and punishment ($\beta^P$). Some models were tested with the inclusion of a lapse term ($\xi$). (**b**) Distributions of individual parameter values from the winning model. (**c**) The winning model was able to reproduce the proportion of conflict option choices over all trials in the observed data with high accuracy (observed vs predicted data r = 0.97). (**d**) The distribution of the reward-punishment sensitivity index – the computational measure of approach-avoidance bias. Higher values indicate approach biases, whereas lower values indicate avoidance biases.

To quantify individual approach-avoidance bias computationally, we calculated the ratio between the reward and punishment sensitivity parameters ($\beta^r/\beta^p$). As the task requires simultaneously balancing reward pursuit (i.e. approach) and punishment avoidance, this composite measure provides an index of where an individual lies on the continuum of approach vs avoidance, with higher and lower values indicating approach and avoidance biases, respectively. We refer to this measure as the 'reward-punishment sensitivity index' (**Figure 3d**).

Comparing parameters across sexes via Welch's t-tests revealed significant differences in reward sensitivity ($t_{289}$ = –2.87, p = 0.004, d = 0.34; lower in females) and consequently reward-punishment sensitivity index ($t_{336}$ = –2.03, p = 0.043, d = 0.22; lower in females, i.e. more avoidance-driven). In the replication sample, we observed the same effect on reward-punishment sensitivity index ($t_{626}$ = –2.79, p = 0.005, d = 0.22; lower in females). However, the sex difference in reward sensitivity did not replicate (p=0.441), although we did observe a significant sex difference in punishment sensitivity in the replication sample ($t_{626}$ = 2.26, p = 0.024, d = 0.18).

## Computational mechanisms of anxiety-related avoidance

We assessed the relationship between task-induced anxiety and individual model parameters through permutation testing of non-parametric correlations, since the distribution of task-induced anxiety was non-Gaussian. The punishment learning rate and reward-punishment sensitivity index were significantly and negatively correlated with task-induced anxiety in the discovery sample (punishment learning rate: Kendall's $\tau$ = –0.088, p = 0.015; reward-punishment sensitivity index: $\tau$ = –0.099, p = 0.005; **Figure 4a and b**). These associations were also significant in the replication sample (punishment learning rate:

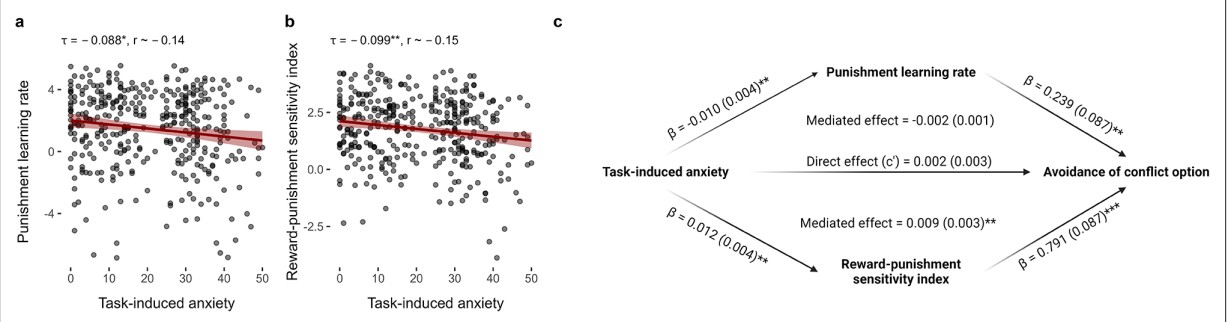

**Figure 4.** Relationships between task-induced anxiety, model parameters, and avoidance. (**a**) Task-induced anxiety was negatively correlated with the punishment learning rate. (**b**) Task-induced anxiety was also negatively correlated with reward-punishment sensitivity index. Kendall's tau correlations and approximate Pearson's *r* equivalents are reported above each figure (n = 369). (**c**) The mediation model. Mediation effects were assessed using structural equation modelling. Bold terms represent variables and arrows depict regression paths in the model. The annotated values next to each arrow show the regression coefficient associated with that path, denoted as *coefficient (standard error)*. Only the reward-punishment sensitivity index significantly mediated the effect of task-induced anxiety on avoidance. Significance levels in all figures are shown according to the following: $p < 0.05$ – *; $p < 0.01$ – **; $p < 0.001$ – ***.

$\tau$ = –0.064, $p$ = 0.020; reward-punishment sensitivity index: $\tau$ = –0.096, $p < 0.001$), with an additional positive correlation between punishment sensitivity and anxiety ($\tau$ = 0.076, $p$ = 0.004; *Appendix 1—table 1*) which may have been detected due to greater statistical power. The learning rate equivalent of the reward-punishment sensitivity index (i.e. $\alpha^r/\alpha^p$) did not significantly correlate with task-induced anxiety (discovery sample: $\tau$ = 0.048, $p$ = 0.165; replication sample: $\tau$ = 0.029, $p$ = 0.298).

Given that both the punishment learning rate and the reward-punishment sensitivity index were significantly and reliably correlated with task-induced anxiety, we next asked whether these computational measures explained the effect of anxiety on avoidance behaviour. We tested this via structural equation modelling, including both the punishment learning rate and reward-punishment sensitivity index as parallel mediators for the effect of task-induced anxiety on proportion of choice for conflict/safe options. This suggested that the mediating effect of the reward-punishment sensitivity index (standardised $\beta$ = –0.009 ± 0.003, $p$ = 0.003) was over four times larger than the mediating effect of the punishment learning rate, which did not reach the threshold as a significant mediator (standardised $\beta$ = –0.002 ± 0.001, $p$ = 0.052; *Figure 4c*). A similar pattern was observed in the replication sample (mediating effect of reward-punishment sensitivity index: standardised $\beta$ = –0.009 ± 0.002, $p$ < 0.001; mediating effect of punishment learning rate: standardised $\beta$ = –0.001 ± 0.0003, $p$ = 0.132; *Appendix 1—table 1*, *Appendix 3—figure 1*).

This analysis reveals two effects: first, that individuals who reported feeling more anxious during the task were slower to update their estimates of punishment probability; and second, more anxious individuals placed a greater weight on the aversive sound *relative to the reward* when making their decisions, meaning that the potential for a reward was more strongly offset by potential for punishment. Critically, the latter but not the former effect explained the effect of task-induced anxiety on avoidance behaviour, thus providing an important insight into the computational mechanisms driving anxiety-related avoidance.

## Psychiatric symptoms predict task-induced anxiety, but not avoidance

We assessed the clinical potential of the measure by investigating how model-agnostic and computational indices of behaviour correlate with self-reported psychiatric symptoms of anxiety, depression, and experiential avoidance; the latter indexes an unwillingness to experience negative emotions/experiences (*Hayes et al., 1996*). We included experiential avoidance as a direct measure of avoidance symptoms/beliefs, given that our task involved avoiding stimuli inducing negative affect. Given that we recruited samples without any inclusion/exclusion criteria relating to mental health, the instances of clinically relevant anxiety symptoms, defined as a score of 10 or greater in the Generalised Anxiety Disorder 7-item scale (*Spitzer et al., 2006*), were low in both the discovery (16%, 60 participants) and the replication sample (20%, 127 participants).

Anxiety, depression, and experiential avoidance symptoms were all significantly and positively correlated with task-induced anxiety ($p < 0.001$; *Appendix 1—table 1*). Only experiential avoidance symptoms were significantly correlated with the proportion of conflict/safe option choices ($\tau = -0.059$, $p = 0.010$; *Appendix 1—table 1*), but this effect was not observed in the replication sample ($\tau = -0.029$, $p = 0.286$). Experiential avoidance was also significantly correlated with punishment learning rates (Kendall's $\tau = -0.09$, $p = 0.024$) and the reward-punishment sensitivity index ($\tau = -0.09$, $p = 0.034$) in the discovery sample. However, these associations were non-significant in the replication sample ($p > 0.3$), and no other significant correlations were detected between the psychiatric symptoms and model parameters (*Appendix 1—table 1*).

## Split-half reliability

We assessed the split-half reliability of the task by correlating the overall proportion of conflict option choices and model parameters from the winning model across the first and second half of trials. For overall choice proportion, reliability was simply calculated via Pearson's correlations. For the model parameters, we calculated model-derived estimates of Pearson's $r$ values from the parameter covariance matrix when first- and second-half parameters were estimated within a single model, following a previous approach recently shown to accurately estimate parameter reliability (*Waltmann et al., 2022*). We interpreted indices of reliability based on conventional values of <0.40 as poor, 0.4–0.6 as fair, 0.6–0.75 as good, and >0.75 as excellent reliability (*Fleiss, 1986*). Overall choice proportion showed good reliability (discovery sample $r = 0.63$; replication sample $r = 0.63$; *Appendix 5—figure 1*). The model parameters showed good-to-excellent reliability (model-derived $r$ values ranging from 0.61 to 0.85 [0.76–0.92 after Spearman-Brown correction]; *Appendix 5—figure 1*).

## Test-retest reliability

We assessed the test-retest reliability of the task by retesting 57 participants from the replication sample at least 10 days after they initially completed the task. The retest version of the task was identical to the initial version, except for the use of new stimuli to represent the conflict and safe options and different latent outcome probabilities to limit practice effects. The test-retest reliability of model-agnostic effects was assessed using the intra-class correlation (ICC) coefficient, specifically ICC(3,1) (following the notation of *Shrout and Fleiss, 1979*). Test-retest reliability of the model parameters was assessed by calculating Pearson's correlations derived from the parameter covariance matrix when parameters from both sessions were estimated within a single model.

Task-induced anxiety, overall proportion of conflict option choices, and the model parameters demonstrated fair to excellent reliability (reliability indices ranging from 0.4 to 0.77; *Appendix 6—figure 1*), except for the punishment learning rate which showed a model-derived reliability of $r = -0.45$. Practice effects analyses showed that task-induced anxiety and the punishment learning rate were also significantly lower in the second session (anxiety: $t_{56} = 2.21$, $p = 0.031$; punishment learning rate: $t_{56} = 2.24$, $p = 0.029$), whilst the overall probability of choosing the conflict option and the other parameters did not significantly change over sessions (*Appendix 7—figure 1*). As the dynamic (latent) outcome probabilities were different across the test vs. retest sessions, in other words participants were not performing the exact same task across sessions, we argue that even 'fair' reliability (i.e. estimates in range of 0.4–0.6) is encouraging, as this suggests that model parameters generalise across different environments if the required computations are preserved. However, potential within-subjects inferences based on the punishment learning rate may not be justified in this task.

## Sensitivity analyses

As our procedure for estimating model parameters (the expectation maximisation (EM) algorithm, see 'Methods') produced high inter-parameter correlations in our data (*Appendix 2—figure 2*), we also re-estimated the parameters using Stan's variational Bayesian inference (VBI) algorithm (*Stan Development Team, 2023*) – this resulted in lower inter-parameter correlations, but our primary computational finding, that the effect of anxiety on choice is mediated by relative sensitivity to reward/punishment was consistent across algorithms (see Appendix 8 for details).

Since participants self-determined the volume of the punishments in the task, and therefore (at least in part) their aversiveness, we also conducted a sensitivity analysis by accounting for self-reported unpleasantness ratings of the punishment (see Appendix 9). Our finding that anxiety impacts

approach-avoidance behaviour was robust to this sensitivity analysis ($p < 0.001$), however the mediating effect of the reward-punishment sensitivity index was not ($p > 0.1$; see Appendix 9 for details). Although sound volume and stimulus unpleasantness correlate (*Seow and Hauser, 2022*), unpleasantness ratings are not a perfect indicator of volume, so this analysis should be interpreted with caution.

## Discussion

To develop a translational and computational model of anxiety-related avoidance, we created a novel task that involves learning under approach-avoidance conflict and investigated the impact of anxiety on task performance. Importantly, we show that individuals who experienced more task-induced anxiety avoid actions that lead to aversive outcomes, at the cost of potential reward. Computational modelling of behaviour revealed that this was explained by relative sensitivity to punishment over reward. However, this effect was limited to task-induced anxiety, and was only evident on a self-report measure of *general* avoidance in our first sample. Finally, split-half and test-retest analyses of the task indicated fair-to-excellent reliability indices of task behaviour and model parameters. These results demonstrate the potential of approach-avoidance reinforcement learning tasks as translational measures of anxiety-related avoidance.

We can evaluate the translational validity of our task based on the criteria of face, construct, predictive, and computational validity (*Willner, 1984*; *Redish et al., 2022*). The task shows sufficient face validity when compared to the Vogel/Geller-Seifter conflict tests; the design involves a series of choices between an action associated with high reward and punishment, or low reward and no punishment, and these contingencies must be learned. One potentially important difference is that our task involved a choice between two active options, referred to as a two-alternative forced choice (2AFC), whereas in the Vogel/Geller-Seifter tests, animals are free to perform an action (e.g. press the lever) or withhold the action (don't press the lever; i.e. a go/no-go design). The disadvantage of the latter approach is that human participants can and typically do show motor response biases (i.e. for either action or inaction), which can confound value-based response biases with approach or avoidance (*Guitart-Masip et al., 2012*). Fortunately, there is ample evidence that rodents can perform 2AFC learning (*Metha et al., 2019*) and conflict tasks (*Simon et al., 2009*; *Oberrauch et al., 2019*; *Glover et al., 2020*). That avoidance responses were positively associated with task-induced anxiety also lends support for the construct validity of our measure, given the theoretical links between anxiety and avoidance under approach-avoidance conflict (*Hayes, 1976*; *Stein and Paulus, 2009*; *Aupperle and Paulus, 2010*). However, we note that the empirical effect size was small ($|r| \sim 0.13$) – we discuss potential reasons for this below.

The predictive and computational validity of the task need to be determined in future work. Predictive validity can be assessed by investigating the effects of anxiolytic drugs on task behaviour. Reduced avoidance after anxiolytic drug administration, or indeed any manipulation designed to reduce anxiety, would support our measure's predictive validity. Similarly, computational validity would also need to be assessed directly in non-human animals by fitting models to their behavioural data. This should be possible even in the face of different procedures across species such as number of trials or outcomes used (shock or aversive sound). We are encouraged by our finding that the winning computational model in our study relies on a relatively simple classical reinforcement learning strategy. There exist many studies showing that non-human animals rely on similar strategies during reward and punishment learning (*Schultz, 2013*; *Mobbs et al., 2020*); albeit to our knowledge this has never been modelled in non-human animals where rewards and punishment can occur simultaneously.

One benefit of designing a task amenable to the computational modelling of behaviour was that we could extract model parameters that represent individual differences in precise cognitive mechanisms. Our model recapitulated individual- and group-level behaviour with high fidelity, and mediation analyses suggested a potential mechanism of how anxiety may induce avoidance, namely through the relative sensitivity to punishment compared to reward. This demonstrates the potential of computational methods in allowing for mechanistic insights into anxiety-related avoidance. Further, we found satisfactory test-retest reliability of the reward-punishment sensitivity index in test-retest analyses, suggesting that the task can be used for within-subject experiments (e.g. on/off medication) to assess the impact of interventions on the mechanisms underlying avoidance.

The composite nature of the reward-punishment sensitivity index may be important for contextualising prior results of anxiety and outcome sensitivity. The traditional view has long been that anxiety is

associated with greater automatic sensitivity to punishments and threats (i.e. that anxious individuals show a negative bias in information processing; *Bishop, 2007*), due to amygdala hypersensitivity (*Mathews and Mackintosh, 1998*; *Hartley and Phelps, 2012*). However, subsequent studies have found inconsistent findings relative to this hypothesis (*Bishop and Gagne, 2018*). We found that, rather than a straightforward link between anxiety and punishment sensitivity, the *relative* sensitivity to punishment over reward provided a better explanation of anxiety's effect on behaviour. This effect is more in keeping with cognitive accounts of anxiety's effect on information processing that involve higher-level cognitive functions such as attention and cognitive control, which propose that anxiety involves a global reallocation of resources away from processing positive and towards negative information (*Beck and Clark, 1997*; *Cisler and Koster, 2010*; *Robinson et al., 2013*), possibly through prefrontal control mechanisms (*Carlisi and Robinson, 2018*). Future studies using experimental manipulations of anxiety will be needed to test the causal role of anxiety on relative sensitivities to punishments versus rewards.

Since we developed our task with the primary focus on translational validity, its design diverges from other reinforcement learning tasks that involve reward and punishment outcomes (*Pike and Robinson, 2022*). One important difference is that we used distinct reinforcers as our reward and punishment outcomes, compared to many studies which use monetary outcomes for both (e.g. earning and losing £1 constitute the reward and punishment, respectively; *Pizzagalli et al., 2005*; *Aylward et al., 2019*; *Jean-Richard-Dit-Bressel et al., 2021*; *Sharp et al., 2022*). Other tasks have been used that induce a conflict between value and motor biases, relying on prepotent biases to approach/move towards rewards and withdraw from punishments, which makes it difficult to approach punishments and withdraw from rewards (*Guitart-Masip et al., 2012*; *Mkrtchian et al., 2017*). However, since translational operant conflict tasks typically induce a conflict between different types of outcome (e.g. food and shocks/sugar and quinine pellets; *van den Bos et al., 2014*; *Oberrauch et al., 2019*), we felt it was important to implement this feature. One study used monetary rewards and shock-based punishments, but also included four options for participants to choose from on each trial, with rewards and punishments associated with all four options (*Seymour et al., 2012*). This effectively requires participants to maintain eight probability estimates (i.e. reward and punishment at each of the four options) to solve the task, which may be too difficult for non-human animals to learn efficiently.

Notably, although a recent computational meta-analysis of reinforcement learning studies showed that symptoms of anxiety and depression are associated with elevated punishment learning rates (*Pike and Robinson, 2022*), we did not observe this pattern in our data. Indeed, we even found the contrary effect in relation to task-induced anxiety, specifically that anxiety was associated with lower rates of learning from punishment. However, other work has suggested that the direction of this effect can depend on the form of anxiety, where cognitive anxiety may be associated with elevated learning rates, but somatic anxiety may show the opposite pattern (*Wise and Dolan, 2020*) and this may explain the discrepancy in findings. Additionally, parameter values are highly dependent on task design (*Eckstein et al., 2022*), and study designs to date may be more optimised in detecting differences in learning rate (*Pike and Robinson, 2022*) – future work is needed to better understand the potentially complex association between anxiety and punishment learning rate. Lastly, as punishment learning rate was severely unreliable in the test-retest analyses, and the associations between punishment learning rate and state anxiety were not robust to an alternative method of parameter estimation (VBI), the negative correlation observed in our study should be treated with caution.

One potentially important limitation of our findings is the small effect size observed in the correlation between task-induced anxiety and avoidance (Kendall's tau values < 0.1, mediation coefficients < 0.01). This may be attributed to the simplicity of using overall choice proportion as a measure of approach/avoidance, as the effect of anxiety on choice was also influenced by punishment probability. This may have also been due to the aversiveness of the punishment outcomes (aversive sounds) used in our study, compared to other studies which included aversive images (*Aupperle et al., 2011*), shocks (*Talmi et al., 2009*), or used virtual reality (*Biedermann et al., 2017*) to provide negative reinforcement. Using more salient punishments would likely produce larger effects. Further, whilst we included tests to check whether participants were wearing headphones for this online study, we could not be certain that the sounds were not played over speakers and/or played at a sufficiently high volume to be aversive. Finally, it may simply be that the relationships between psychiatric symptoms and behaviour are small due to the inherent variability of individuals, the lack of precision of

self-reported symptoms, or the unreliability of mental health diagnoses (*Freedman et al., 2013*; *Wise et al., 2023*). Despite these limitations, an advantage of using web-based auditory stimuli is their efficiency in scaling with sample size, as only a web-browser is needed to complete the task. This is especially relevant for psychiatric research, given the need for studies with larger sample sizes in the field (*Rutledge et al., 2019*).

Relatedly, participants had some control over the intensity at which the punishments were presented, which may have driven our findings relating to anxiety and putative mechanisms of anxiety-related avoidance. Sensitivity analyses showed that our finding that anxiety is positively associated with avoidance in the task was robust to individual differences in self-reported punishment unpleasantness, whilst the mediation effects were not. Future work imposing better control over the stimuli presented, and/or using within-subjects designs will be needed to validate the role of reward/punishment sensitivities in anxiety-related avoidance.

The punishment learning rate was severely unreliable given a model-derived correlation across sessions of $r = -0.45$, whereas the other indices of task performance showed fair-to-excellent reliability. This might be a result of practice effects, where participants learned about the punishments differently across sessions, but not about the rewards. Alternatively, as participants self-determined the loudness of the punishments, differences in volume settings across sessions may have impacted the reliability of this parameter (and indeed other measures). Further, the asymmetric nature of the task may have impacted our ability to estimate the punishment learning rate, as there were fewer occurrences of the punishment compared to the reward. However, given that the sensitivity parameters were more important for explaining how anxiety affected avoidance, the low observed reliability of the punishment learning rate may not present a barrier for future use of this task in repeated-measures designs.

We also did not find any significant correlations between task performance and clinical symptoms of anxiety, which raises questions around the predictive validity of the task. As participants were recruited without imposing specific inclusion/exclusion criteria related to mental health, the low level of clinically relevant symptoms in the data may have contributed to these results. Further studies in samples with high levels of anxiety or examining individuals at the extreme ends of symptom distributions may offer increased sensitivity in assessing the predictive validity of the task. Further, it is worth noting that many animal paradigms were developed and widely adopted due to their sensitivity to anxiolytic medication (*Cryan and Holmes, 2005*). Given the lack of associations with clinical measures in our results, it is possible that current translational models of anxiety may not fully capture behaviours that are directly relevant to pathological anxiety. To develop translational paradigms of clinical utility, future research should place a stronger emphasis on assessing their clinical validity in humans. Using multi-level or continuous outcomes would also improve the ecological validity of the present approach and interpretation of the sensitivity indices.

Finally, whilst there is a broad literature on the roles of behavioural inhibition and avoidance tendency traits on decision-making and behaviour (*Gray, 1982*; *Carver and White, 1994*; *Corr, 2004*), we did not replicate the correlation of experiential avoidance and avoidance responses or the reward-punishment sensitivity index. Since there were also no significant correlations across task performance indices and clinical symptom measures, our findings suggest that the measure may be more sensitive to behaviours relating to state anxiety, rather more stable traits. Nevertheless, how performance in the present task relates to other traits such as behavioural approach/inhibition tendencies (*Carver and White, 1994*), as has been found in previous studies on reward/punishment learning (*Wise and Dolan, 2020*; *Sharp et al., 2022*) and approach-avoidance conflict (*Aupperle et al., 2011*), will be an important question for future work.

## Conclusion

Avoidance is a hallmark symptom of anxiety-related disorders and there is a need for better translational models of avoidance. In our novel translational task designed to involve the same computational processes as those involved in non-human animal tests, anxiety was reliably induced, associated with avoidance behaviour, and linked to a computational model of behaviour. We hope that similar approaches will facilitate progress in the theoretical understanding of avoidance and ultimately aid in the development of novel anxiolytic treatments for anxiety-related disorders.

## Methods

### Participants

We recruited participants from Prolific (https://www.prolific.com/). Participants had to be aged 18–60, speak English as their first language, have no language-related disorders/literacy difficulties, have no vision/hearing difficulties/have no mild cognitive impairment or dementia, and be resident in the UK. Participants were reimbursed at a rate of £7.50 per hour, and could earn a bonus of up to an additional £7.50 per hour based on their performance on the task. The study had ethical approval from the University College London Research Ethics Committee (ID 15253/001).

We first recruited a discovery sample to investigate whether anxiety was associated with avoidance in the novel task. We conducted an a priori power analysis to detect a minimally interesting correlation between anxiety and avoidance of $r = 0.15$ ($alpha_{two-tailed}=0.05$, power = 0.80), resulting in a required sample size of 346. We aimed to replicate our initial finding (that task-induced anxiety was associated with the model-derived measure of approach-avoidance bias) in a replication sample. In the initial sample, the size of this effect was Kendall's $\tau = 0.099$, which approximates to a Pearson's $r$ of 0.15 – this was the basis for the power analysis for the replication sample. Therefore, the power analysis was configured to detect a correlation of $r = 0.15$ ($alpha_{one-tailed}=0.05$, power = 0.95), resulting in a required sample size of n = 476. A total of 423 participants completed the initial study (mean age = 38, SD = 10, 59% female), and 726 participants completed the replication study (mean age = 34, SD = 10, 51% female).

### Headphone screen and calibration

We used aversive auditory stimuli as our punishments in the reinforcement learning task, which consisted of combinations of female screams and high-frequency tones that were individually rated as highly aversive in a previous online study (*Seow and Hauser, 2022*). The stimuli can be found online at https://osf.io/m3zev/. To maximise their aversiveness, we asked participants to use headphones whilst they completed the task. All participants completed an open-source, validated auditory discrimination task (*Woods et al., 2017*) to screen out those who were unlikely to be using headphones. Participants could only continue to the main task if they achieved an accuracy threshold of five out of six trials in this screening task.

To further increase the aversiveness of the punishments, we wanted each participant to hear them at a loud volume. We initially used a calibration task to increase the system volume and implemented attention checks in the main task to ensure that participants had kept the volume at a sufficiently high level during the task (see below). Before beginning the calibration task, participants were advised to set their system volume to around 10% of the maximum possible. Then, we presented two alternating auditory stimuli: a burst of white noise which was set to have the same volume as the aversive sounds in the main task, and a spoken number played at a lower volume to one ear ('one', 'two', or 'three'; these stimuli are also available online at https://osf.io/m3zev/). We instructed participants to adjust their system volume so that the white noise was loud but comfortable. At the same time, they also had to press the number key on their keyboard corresponding to the number spoken between each white noise presentation. The rationale for the difference in volume was to fix a lower boundary for each participant's system volume, so that the volume of the punishments presented in the main task was locked above this boundary. We used the white noise as a placeholder for the punishments to limit desensitisation. Since the volume of the punishments, and therefore at least in part their aversiveness, was determined by the participants themselves, we conducted sensitivity analyses on all reported findings after accounting for self-reported unpleasantness ratings for the punishment – this did not materially affect our main findings (see Appendix 9).

### The approach-avoidance reinforcement learning task

We developed a novel task on which participants learned to simultaneously pursue rewards and avoid punishments. On each trial, participants had 2 seconds to choose one of two options, depicted as distinct visual stimuli (*Figure 2a*), using the left/right arrow keys on their keyboard. The chosen option was then highlighted by a white border, and participants were presented with the outcome for that trial for 2 seconds. This could be one of (1) no outcome, (2) a reward (an animated coin image), (3) a punishment (an aversive sound), or (4) both the reward and punishment. We incentivised participants to collect the coins by informing them that they could earn a bonus payment based on the number

of coins earned during the task. If no choice was made, the text 'Too slow!' was shown for 2 seconds and the trial was skipped. Participants completed 200 trials which took 10 minutes on average across the sample. Each option was associated with separate probabilities of producing the reward/aversive sound outcomes (*Figure 2b*), and these probabilities fluctuated randomly over trials. Both options were associated with the reward, but only one was associated with the punishment (i.e. the 'conflict' option could produce both rewards and punishments, but the 'safe' option could produce only rewards and never punishments). When asked to rate the unpleasantness of the punishments from 'Not at all' to 'Extremely' (scored from 0 to 50, respectively), participants gave a mean rating of 31.7 (SD = 12.8).

The latent outcome probabilities were generated as decaying Gaussian random walks in a similar fashion to previous restless bandit tasks (*Daw et al., 2006*), where the probability, p, of observing outcome, *o*, at option, *i*, on trial, *t*, drifted according to the following:

$$p_{i,o}\left(t+1\right) = \lambda p_{i,o}\left(t\right) + \left(1-\lambda\right)\theta + e$$

$$e \sim N\left(0, 0.04^2\right)$$

The decay parameter, $\lambda$, was set to 0.97, which pulled all values towards the decay centre $\theta$, which was set to 0.5. The random noise, *e*, added on each trial was drawn from a zero-mean Gaussian with a standard deviation of 0.04. All probabilities of observing the reward at the safe option were scaled down by a factor of 0.7 so that it was less rewarding than the conflict option, on average. We initialised each walk at the following values: conflict option reward probability – 0.7, safe option reward probability – 0.35, conflict option punishment probability – 0.2. We only used sets of latent outcome probabilities where the cross-correlations across each probability walk were below 0.3. A total of four sets of outcome probabilities were used across our studies – they are available online at https://osf.io/m3zev/.

We implemented three auditory attention checks during the main task, to ensure that participants were still using an appropriate volume to listen to the auditory stimuli. As participants could potentially mute their device during the outcome phase of the trial only (i.e. only when the aversive sounds could occur), the timing of these checks had to coincide with moment of outcome presentation. Therefore, we implemented these checks at the outcome phase of two randomly selected trials where the participants' choices led to no outcome (no reward or punishment), and also once at the very end of the task. Specifically, we played one of the spoken numbers used during the calibration task (see above) and the participant had 2.5 seconds to respond using the corresponding number key on their keyboard.

Immediately after the task and the final attention check, we obtained subjective ratings about the task using the following questions: 'How unpleasant did you find the sounds?' and 'How anxious did you feel during the task?'. Responses were measured on a continuous slider ranging from 'Not at all' (0) to 'Extremely' (50).

## Data cleaning

As the task was implemented online where we could not ensure the same testing standards as we could in-person, we used three exclusion criteria to improve data quality. Firstly, we excluded participants who missed a response to more than one auditory attention check (see above; 8% in both discovery and replication samples) – as these occurred infrequently and the stimuli used for the checks were played at relatively low volume, we allowed for incorrect responses so long as a response was made. Secondly, we excluded those who responded with the same response key on 20 or more consecutive trials (>10% of all trials; 4%/6% in discovery and replication samples, respectively) – note that as the options randomly switched sides on the screen across trials, this did not exclude participants who frequently and consecutively chose a certain option. Lastly, we excluded those who did not respond on 20 or more trials (1%/2% in discovery and replication samples, respectively). Overall, we excluded 51 out of 423 (12%) in the discovery sample, and 98 out of 725 (14%) in the replication sample. We conducted all the analyses with and without these exclusions. These exclusions had no effects on the main findings reported in the manuscript, and differences before and after excluding data are reported in Appendix 1.

**Table 1.** Model specification.

| Model | Parameters | | | |
|---|---|---|---|---|
| Reward learning only | $\alpha^r$ | | $\beta^r$ | |
| Punishment learning only | $\alpha^p$ | | $\beta^p$ | |
| Symmetrical learning | $\alpha$ | | $\beta$ | |
| Asymmetric learning rates | $\alpha^r$ | $\alpha^p$ | $\beta$ | |
| Asymmetric sensitivities | $\alpha$ | | $\beta^r$ | $\beta^p$ |
| Asymmetric learning rates and sensitivities | $\alpha^r$ | $\alpha^p$ | $\beta^r$ | $\beta^p$ |

All custom model scripts are available online at https://osf.io/m3zev/.

## Mixed-effects regressions of trial-by-trial choice

We specified two hierarchical logistic regression models to test the effect of the dynamic outcome probabilities on choice. Both models included predictors aligned with approach and avoidance responses, based on the latent probabilities of observing an outcome given the chosen option at trial $t$, P(outcome | option)$_t$. First, if individuals were learning to choose maximise their rewards earned, they should generally be choosing the option which was relatively more likely to produce the reward on each trial. This would result in a significant effect of the difference in reward probabilities across options, as given by: $\delta$P(reward)$_t$ = P(reward | conflict)$_t$ - P(reward | safe)$_t$. Second, if individuals were learning to avoid punishments, they should generally be avoiding the conflict option on trials when it is likely to produce the punishment. This would result in a significant effect of the punishment probability at the conflict option: P(punishment | conflict)$_t$.

The first regression model simply included these two predictors and random intercepts varying by participant to predict choice:

$$\text{Choice} \sim \delta\text{P(reward)} + \text{P(punishment | conflict)} + (1 \mid \text{participant})$$

The second model tested the effect of task-induced anxiety on choice, by including anxiety as a main effect and its interactions with $\delta$P(reward)$_t$ and P(punishment | conflict)$_t$:

$$\text{Choice} \sim \delta\text{P(reward)} + \text{P(punishment | conflict)} + \text{Anxiety} + \text{Anxiety} \cdot \delta\text{P(reward)} + \text{Anxiety} \cdot$$
$$\text{P(punishment | conflict)} + (1 \mid \text{participant})$$

## Computational modelling using reinforcement learning

### Model specification

We used variations of classical reinforcement learning algorithms (*Sutton and Barto, 2018*), specifically the 'Q-learning' algorithm (*Watkins and Dayan, 1992*), to model participants' choices in the main task (*Table 1*). Briefly, our models assume that participants estimate the probabilities of observing a reward and punishment separately for each punishment, and continually update these estimates after observing the outcomes of their actions. Consequently, participants can then use these estimates to choose the option which will maximise subjective value, for example by choosing the option which is more likely to produce a reward, or avoiding an option if it is very likely to produce a punishment. By convention, we refer to the probability estimates as 'Q-values'; for example, the estimated probability of observing a reward after choosing the conflict option is written as $Q^r$ (conflict), whereas the probability of observing a punishment after choosing the safe option would be written as $Q^p$ (safe).

In all models, the probability estimates of observing outcome, $o = \{r, p\}$, of the chosen option, $a = \{\text{conflict}, \text{safe}\}$, on trial, $t$, were updated via: $Q_{t+1}^o (a) = Q_t^o (a) + \alpha \cdot [o^t - Q_t^o (a)]$, where the learning rate is controlled by $\alpha \in (0, 1)$. In some models (*Table 1*), the learning rate was split into reward- and punishment-specific parameters, $\alpha^r$ and $\alpha^p$.

On each trial, the probability estimates for observing rewards and aversive sounds are integrated into action weights, $W$, via:

$W = \beta \cdot (Q^r - Q^p)$, where individual differences in sensitivity to the outcomes (how much the outcomes affect value-based choice) are parameterised by the outcome sensitivity parameter, $\beta$. In models with a single sensitivity parameter, participants were assumed to value the reward and punishment equally, such that obtaining a reward had the same subjective value as avoiding a punishment. However, as individual differences in approach-avoidance conflict could manifest in asymmetries in the valuation of reward/punishment, we also specified models (*Table 1*) where $\beta$ was split into reward- and punishment-specific parameters, $\beta^r$ and $\beta^p$ :

$$W = \beta^r \cdot Q^r - \beta^p \cdot Q^p.$$

The action weights are then the basis on which a choice was made between the options, which was modelled with the softmax function:

$$P(a) = \frac{W(a)}{\sum_i W(i)}.$$

We also tested all models with the inclusion of a lapse term, which allows for choice stochasticity to vary across participants, but since this did not improve model fit for any model, we do not discuss this further.

## Model fitting and comparison

We fit the models using a hierarchical EM algorithm (*Huys et al., 2011*) – the algorithm and supporting code is available online at https://www.quentinhuys.com/pub/emfit/. All prior distributions of the untransformed parameter values were set as Gaussian distributions to regularise fitting and to limit extreme values. Parameters were transformed within the models, via the sigmoid function for parameters constrained to values between 0 and 1; or exponentiated for strictly positive parameters. We compared evidence across models using integrated Bayesian information criterion (*Huys et al., 2011*) which integrates both model likelihood and model complexity into a single measure for model comparison.

## Mediation analysis of anxiety-related avoidance

To investigate the cognitive processes underlying anxiety-related avoidance, we tested for potential mediators of the effect of task-induced anxiety on avoidance choices in the approach-avoidance reinforcement learning task. Mediation was assessed using structural equation modelling with the 'lavaan' package in R (*Rosseel, 2012*). This allowed us to test multiple potential mediators within a single model. We used maximum-likelihood estimation with and without bootstrapped standard errors (using 2000 bootstrap samples) for estimation. The potential mediators were any parameters from our computational model that significantly correlated with task-induced anxiety. Therefore, the model included the following variables: task-induced anxiety, avoidance choices (computed as the proportion of safe option choices for each participant), and model parameters which correlated with task-induced anxiety. The model was constructed such that the model parameters were parallel mediators of the effect of task-induced anxiety on avoidance choices. The model parameters were allowed to covary with one another.

## Symptom questionnaires

Psychiatric symptoms were measured using the Generalised Anxiety Disorder scale (GAD7; *Spitzer et al., 2006*), Patient Health Questionnaire depression scale (PHQ8; *Kroenke et al., 2009*), and the Brief Experiential Avoidance Questionnaire (BEAQ; *Gámez et al., 2014*).

## Test-retest reliability analysis

We retested a subset of our sample (n = 57) to assess the reliability of our task. We determined our sample size via an a priori power analysis using GPower (*Faul et al., 2007*), which was set to detect the smallest meaningful effect size of $r = 0.4$, as any lower test-retest correlations are considered poor by convention (*Fleiss, 1986*; *Cicchetti, 1994*). The required sample size was thus 50 participants, based on a one-tailed significance threshold of 0.05 and 90% power. Participants were recruited from Prolific. We invited participants from the replication study (providing they had not been excluded on

the basis of data quality) to complete the study again, at least 10 days after their first session. The retest version of the approach-avoidance reinforcement learning task was identical to that in the first session, except for the use of new stimuli to represent the conflict and safe options and different latent outcome probabilities to limit practice effects.

We calculated ICCs coefficients to assess the reliability of our model-agnostic measures. Specifically, we used consistency-based ICC(3,1) (following the notation of *Shrout and Fleiss, 1979*). To estimate the reliability of the model-derived measures, we estimated individual- and session-level model parameters via a joint model in an approach recently shown to improve the accuracy of estimating reliability (*Waltmann et al., 2022*). Briefly, this involved fitting behavioural data from both test and retest sessions per participant in a single model, but specifying unique parameters per session. Since our fitting algorithm assumes parameters are drawn from a multivariate Gaussian distribution, information about the reliability of the parameters can be derived from the covariance of each parameter across time. These covariance values can subsequently be converted to Pearson's $r$ values, and thus indices of reliability.

## Materials availability statement

The data and code required to replicate the data analyses are available online at https://osf.io/m3zev/.

## Additional information

### Competing interests

Oliver J Robinson: OJR's MRC senior fellowship is partially in collaboration with Cambridge Cognition Ltd (who plan to provide in-kind contribution) and he is running an investigator-initiated trial with medication donated by Lundbeck (escitalopram and placebo, no financial contribution). He also holds an MRC-Proximity to discovery award with Roche (who provide in-kind contributions and have sponsored travel for ACP) regarding work on heart-rate variability and anxiety. He also has completed consultancy work for Peak, IESO digital health, Roche and BlackThorn therapeutics. OJR sat on the committee of the British Association of Psychopharmacology until 2022. Jonathan P Roiser: Senior editor, eLife. The other author declares that no competing interests exist.

### Funding

| Funder | Grant reference number | Author |
|---|---|---|
| Wellcome Trust | PhD Studentship 222268/Z/20/Z | Yumeya Yamamori |
| Medical Research Council | Senior Non-Clinical Fellowship MR/R020817/1 | Oliver J Robinson |
| Wellcome Trust | Investigator Award 101798/Z/13/Z | Jonathan P Roiser |

The funders had no role in study design, data collection and interpretation, or the decision to submit the work for publication. For the purpose of Open Access, the authors have applied a CC BY public copyright license to any Author Accepted Manuscript version arising from this submission.

### Author contributions

Yumeya Yamamori, Conceptualization, Resources, Data curation, Software, Formal analysis, Funding acquisition, Visualization, Methodology, Writing – original draft, Writing – review and editing; Oliver J Robinson, Jonathan P Roiser, Conceptualization, Supervision, Funding acquisition, Methodology, Writing – original draft, Writing – review and editing

### Author ORCIDs

Yumeya Yamamori  https://orcid.org/0000-0001-5508-7965
Oliver J Robinson  https://orcid.org/0000-0002-3100-1132
Jonathan P Roiser  http://orcid.org/0000-0001-8269-1228

## Ethics

Informed consent and consent to publish was obtained from our participants. The study had ethical approval from the University College London Research Ethics Committee (ID 15253/001).

Reviewer #1 (Public Review): https://doi.org/10.7554/eLife.87720.4.sa1
Reviewer #2 (Public Review): https://doi.org/10.7554/eLife.87720.4.sa2
Reviewer #3 (Public Review): https://doi.org/10.7554/eLife.87720.4.sa3
Author Response https://doi.org/10.7554/eLife.87720.4.sa4

## Additional files

### Supplementary files
• MDAR checklist

### Data availability

The data and code required to replicate the data analyses and visualisations are available online at https://osf.io/m3zev/.

The following dataset was generated:

| Author(s) | Year | Dataset title | Dataset URL | Database and Identifier |
| --- | --- | --- | --- | --- |
| Yamamori Y, Robinson OJ, Roiser JP | 2023 | Data from: Approach-avoidance reinforcement learning as a translational and computational model of anxiety-related avoidance | https://osf.io/m3zev/ | Open Science Framework, m3zev |

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

# Appendix 1

## Comparison of results in the discovery and replication samples, with and without data cleaning exclusions

**Appendix 1—table 1.** Statistical results across the discovery and replication samples, and the effect of data cleaning exclusions.

| Statistical test/model | Sub-test | Sample (*without data cleaning*) Discovery | Discovery |
|---|---|---|---|
| Hierarchical logistic regression; no task-induced anxiety | Reward coefficient | $\beta = 0.98 \pm 0.03$, p < 0.001 | $\beta = 0.95 \pm 0.02$, p < 0.001 |
| | | *$\beta = 0.95 \pm 0.03$, p < 0.001* | *$\beta = 0.90 \pm 0.02$, p < 0.001* |
| | Punishment coefficient | $\beta = -0.33 \pm 0.04$, p < 0.001 | $\beta = -0.52 \pm 0.03$, p < 0.001 |
| | | *$\beta = -0.29 \pm 0.04$, p < 0.001* | *$\beta = -0.50 \pm 0.03$, p < 0.001* |
| Distribution; task-induced anxiety | | Mean = 21, SD = 14 | Mean = 22, SD = 13 |
| | | *Mean = 21, SD = 14* | *Mean = 22, SD = 14* |
| Correlation; task-induced anxiety and choice | | $\tau = -0.074$, p = 0.033 | $\tau = -0.075$, p = 0.005 |
| | | *$\tau = -0.068$, p = 0.036* | *$\tau = -0.081$, p = 0.002* |
| Hierarchical logistic regression; with task-induced anxiety | Reward coefficient | $\beta = 0.98 \pm 0.03$, p < 0.001 | $\beta = 0.95 \pm 0.02$, p < 0.001 |
| | | *$\beta = 0.95 \pm 0.03$, p < 0.001* | *$\beta = 0.90 \pm 0.02$, p < 0.001* |
| | Punishment coefficient | $\beta = -0.33 \pm 0.04$, p < 0.001 | $\beta = -0.52 \pm 0.03$, p < 0.001 |
| | | *$\beta = -0.29 \pm 0.04$, p < 0.001* | *$\beta = -0.50 \pm 0.03$, p < 0.001* |
| | Task-induced anxiety coefficient | $\beta = -0.04 \pm 0.04$, p = 0.304 | $\beta = 0.02 \pm 0.03$, p = 0.637 |
| | | *$\beta = -0.03 \pm 0.04$, p = 0.377* | *$\beta = 0.003 \pm 0.03$, p = 0.925* |
| | Interaction of task-induced anxiety and reward | $\beta = -0.06 \pm 0.03$, p = 0.074 | $\beta = -0.003 \pm 0.02$, p = 0.901 |
| | | *$\beta = -0.05 \pm 0.03$, p = 0.076* | *$\beta = -0.006 \pm 0.02$, p = 0.774* |
| | Interaction of task-induced anxiety and punishment | $\beta = -0.10 \pm 0.04$, p = 0.022 | $\beta = -0.23 \pm 0.03$, p < 0.001 |
| | | *$\beta = -0.10 \pm 0.04$, p = 0.011* | *$\beta = -0.23 \pm 0.03$, p < 0.001* |
| Hierarchical logistic regression; with task-induced anxiety, no interaction | Reward coefficient | $\beta = 0.99 \pm 0.03$, p < 0.001 | $\beta = 0.95 \pm 0.02$, p < 0.001 |
| | | *$\beta = 0.95 \pm 0.03$, p < 0.001* | *$\beta = 0.89 \pm 0.02$, p < 0.001* |
| | Punishment coefficient | $\beta = -0.33 \pm 0.04$, p < 0.001 | $\beta = -0.52 \pm 0.03$, p < 0.001 |
| | | *$\beta = -0.29 \pm 0.04$, p < 0.001* | *$\beta = -0.50 \pm 0.03$, p < 0.001* |
| | Task-induced anxiety coefficient | $\beta = -0.09 \pm 0.03$, p = 0.012 | $\beta = -0.08 \pm 0.03$, p = 0.005 |
| | | *$\beta = -0.08 \pm 0.03$, p = 0.016* | *$\beta = -0.10 \pm 0.03$, p < 0.001* |
| Computational model comparison | Winning model | 2 learning rate, 2 sensitivity | 2 learning rate, 2 sensitivity |
| | | *2 learning rate, 2 sensitivity* | *2 learning rate, 2 sensitivity* |

*Appendix 1—table 1 Continued on next page*

*Appendix 1—table 1 Continued*

| Statistical test/model | Sub-test | Sample (*without data cleaning*) | |
|---|---|---|---|
| | | Discovery | Discovery |
| | | $\tau$ = –0.019, p = 0.596 | $\tau$ = –0.010, p = 0.749 |
| | Reward learning rate | *$\tau$ = –0.006, p = 0.847* | *$\tau$ = –0.012, p = 0.637* |
| | | $\tau$ = –0.088, p = 0.015 | $\tau$ = –0.064, p = 0.019 |
| | Punishment learning rate | *$\tau$ = –0.073, p = 0.026* | *$\tau$ = –0.074, p = 0.003* |
| | | $\tau$ = 0.048, p = 0.175 | $\tau$ = 0.029, p = 0.282 |
| | Learning rate ratio | *$\tau$ = –0.037, p = 0.276* | *$\tau$ = 0.045, p = 0.072* |
| | | $\tau$ = –0.038, p = 0.286 | $\tau$ = –0.028, p = 0.285 |
| | Reward sensitivity | *$\tau$ = –0.047, p = 0.149* | *$\tau$ = –0.023, p = 0.345* |
| | | $\tau$ = 0.068, p = 0.051 | $\tau$ = 0.076, p = 0.004* |
| | Punishment sensitivity | *$\tau$ = 0.047, p = 0.153* | *$\tau$ = 0.094, p < 0.001* |
| Correlation; task-induced anxiety and model parameters | Reward-punishment sensitivity index | $\tau$ = –0.099, p = 0.005 | $\tau$ = –0.096, p < 0.001 |
| | | *$\tau$ = –0.084, p = 0.011* | *$\tau$ = –0.103, p < 0.001* |
| | Punishment learning rate | $\beta$ = –0.002 ± 0.001, p = 0.052 | $\beta$ = –0.001 ± 0.003, p = 0.132 |
| | | *$\beta$ = –0.001 ± 0.001, p = 0.222* | *$\beta$ = –0.001 ± 0.003, p = 0.031†* |
| Mediation | Reward-punishment sensitivity index | $\beta$ = 0.009 ± 0.003, p = 0.003 | $\beta$ = 0.009 ± 0.002, p < 0.001 |
| | | *$\beta$ = 0.006 ± 0.002, p = 0.011* | *$\beta$ = 0.009 ± 0.002, p < 0.001* |
| | GAD7 | $\tau$ = –0.026, p = 0.458 | $\tau$ = –0.001, p = 0.988 |
| | | *$\tau$ = 0.005, p = 0.894* | *$\tau$ = –0.002, p = 0.919* |
| | PHQ8 | $\tau$ = –0.02, p = 0.579 | $\tau$ = –0.013, p = 0.639 |
| | | *$\tau$ = 0.014, p = 0.684* | *$\tau$ = –0.012, p = 0.646* |
| Correlation; psychiatric symptoms and choice (overall proportion of conflict option choices) | BEAQ | $\tau$ = –0.059, p = 0.010 | $\tau$ = –0.029, p = 0.286* |
| | | *$\tau$ = –0.048, p = 0.151†* | *$\tau$ = –0.020, p = 0.423* |
| | GAD7 | $\tau$ = 0.256, p < 0.001 | $\tau$ = 0.222, p < 0.001 |
| | | *$\tau$ = 0.267, p < 0.001* | *$\tau$ = 0.231, p < 0.001* |
| | PHQ8 | $\tau$ = 0.233, p < 0.001 | $\tau$ = 0.184, p < 0.001 |
| | | *$\tau$ = 0.244, p < 0.001* | *$\tau$ = 0.194, p < 0.001* |
| Correlation; psychiatric symptoms and task-induced anxiety | BEAQ | $\tau$ = 0.15, p < 0.001 | $\tau$ = 0.176, p < 0.001 |
| | | *$\tau$ = 0.172, p < 0.001* | *$\tau$ = 0.17, p < 0.001* |

*Appendix 1—table 1 Continued on next page*

*Appendix 1—table 1 Continued*

| Statistical test/model | Sub-test | Sample (*without data cleaning*) | |
| --- | --- | --- | --- |
| | | **Discovery** | **Discovery** |
| | | *τ* = –0.06, p = 0.076 | *τ* = –0.03, p = 0.304 |
| | Reward learning rate | *τ = –0.061, p = 0.074* | *τ = –0.028, p = 0.264* |
| | | *τ* = –0.07, p = 0.077 | *τ* = –0.01, p = 0.621 |
| | Punishment learning rate | *τ = –0.037, p = 0.265* | *τ = –0.017, p = 0.534* |
| | | *τ* = –0.001, p = 0.999 | *τ* = –0.034, p = 0.229 |
| | Learning rate ratio | *τ = –0.032, p = 0.35* | *τ = –0.026, p = 0.301* |
| | | *τ* = –0.01, p = 0.802 | *τ* = –0.01, p = 0.745 |
| | Reward sensitivity | *τ = –0.005, p = 0.877* | *τ = –0.009, p = 0.718* |
| | | *τ* = 0.05, p = 0.154 | *τ* = 0.003, p = 0.907 |
| | Punishment sensitivity | *τ = 0.022, p = 0.513* | *τ = 0.012, p = 0.633* |
| Correlation; GAD7 and model parameters | | *τ* = –0.05, p = 0.149 | *τ* = 0.01, p = 0.746 |
| | Reward-punishment sensitivity index | *τ = –0.023, p = 0.496* | *τ = 0.007, p = 0.797* |
| | | *τ* = –0.03, p = 0.396 | *τ* = –0.03, p = 0.219 |
| | Reward learning rate | *τ = –0.04, p = 0.239* | *τ = –0.035, p = 0.17* |
| | | *τ* = –0.06, p = 0.100 | *τ* = –0.01, p = 0.630 |
| | Punishment learning rate | *τ = –0.022, p = 0.504* | *τ = –0.025, p = 0.348* |
| | | *τ* = 0.008, p = 0.809 | *τ* = –0.033, p = 0.231 |
| | Learning rate ratio | *τ = –0.038, p = 0.259* | *τ = –0.016, p = 0.521* |
| | | *τ* = –0.012, p = 0.610 | *τ* = 0.01, p = 0.729 |
| | Reward sensitivity | *τ = –0.009, p = 0.801* | *τ = 0.004, p = 0.901* |
| | | *τ* = 0.05, p = 0.179 | *τ* = –0.002, p = 0.948 |
| | Punishment sensitivity | *τ = 0.008, p = 0.802* | *τ = 0.012, p = 0.619* |
| Correlation; PHQ8 and model parameters | | *τ* = –0.06, p = 0.123 | *τ* = 0.02, p = 0.557 |
| | Reward-punishment sensitivity index | *τ = –0.01, p = 0.773* | *τ = 0.005, p = 0.867* |
| | | *τ* = –0.06, p = 0.085 | *τ* = –0.02, p = 0.394 |
| | Reward learning rate | *τ = –0.047, p = 0.147* | *τ = –0.017, p = 0.503* |
| | | *τ* = –0.08, p = 0.024 | *τ* = –0.03, p = 0.337* |
| | Punishment learning rate | *τ = –0.071, p = 0.032* | *τ = –0.031, p = 0.228* |
| | | *τ* = 0.018, p = 0.618 | *τ* = –0.005, p = 0.883 |
| | Learning rate ratio | *τ = 0.002, p = 0.938* | *τ = 0.007, p = 0.759* |
| | | *τ* = –0.01, p = 0.739 | *τ* = 0.01, p = 0.753 |
| | Reward sensitivity | *τ = –0.036, p = 0.29* | *τ = 0.002, p = 0.927* |
| | | *τ* = 0.07, p = 0.061 | *τ* = 0.02, p = 0.477 |
| | Punishment sensitivity | *τ = 0.05, p = 0.127* | *τ = 0.025, p = 0.308* |
| Correlation; BEAQ and model parameters | | *τ* = 0.02, p = 0.034 | *τ* = –0.01, p = 0.745* |
| | Reward-punishment sensitivity index | *τ = – 0.073, p = 0.028* | *τ = – 0.015, p = 0.548* |

| | | Sample (*without data cleaning*) | |
|---|---|---|---|
| Statistical test/model | Sub-test | Discovery | Discovery |

Note. All correlations tests were conducted using permutation-based Kendall's tau correlations, using 10,000 permutations per test. Symbols represent the following.

*Inconsistent findings across discovery replication samples.

†Inconsistent findings before and after data cleaning exclusions. Abbreviations: GAD7 – Generalised Anxiety Disorder 7-item scale; PHQ8 – Patient Health Questionnaire 8-item depression scale; BEAQ – Brief Experiential Avoidance Questionnaire.

A sensitivity analysis of the effect of implementing or not implementing our data cleaning exclusions showed that the significant and negative correlation between experiential avoidance symptoms and overall proportion of conflict option choices ($\tau = -0.059$, $p = 0.010$) was not significant when the data cleaning exclusions were not implemented ($\tau = -0.029$, $p = 0.286$), however since this effect was not replicated in the independent sample (with and without data exclusions), we do not discuss this further. The mediating effect of the punishment learning rate was also significant in the replication sample only before excluding data, however the effect sizes were small and similar before and after excluding data, and so we also do not discuss this further.

## Appendix 2

### Computational modelling

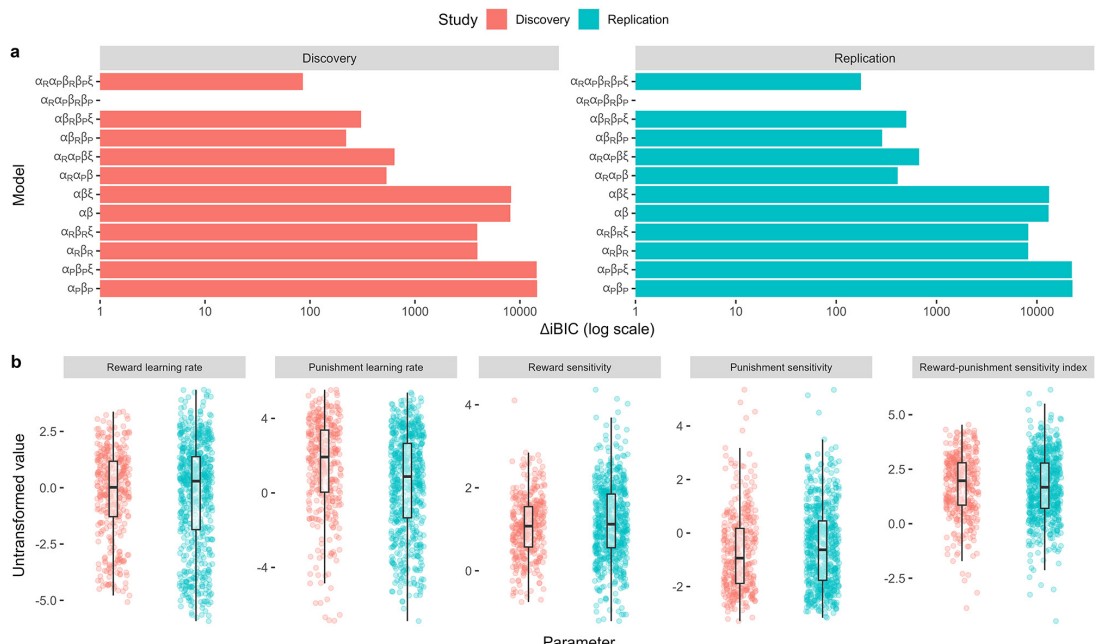

**Appendix 2—figure 1.** Model comparison and parameter distributions across studies. (**a**) Model comparison results. The difference in integrated Bayesian information criterion scores from each model relative to the winning model is indicated on the x-axis. The winning model in both studies included specific learning rates for reward ($\alpha^R$) and punishment learning ($\alpha^p$), and specific outcome sensitivity parameters for reward ($\beta^R$) and punishment ($\beta^P$). Some models were tested with the inclusion of a lapse term ($\xi$). (**b**) Distributions of individual parameter values from the winning model across studies. The reward-punishment sensitivity index constituted our computational measure of approach-avoidance bias, calculated by taking the ratio between the reward and punishment sensitivity parameters.

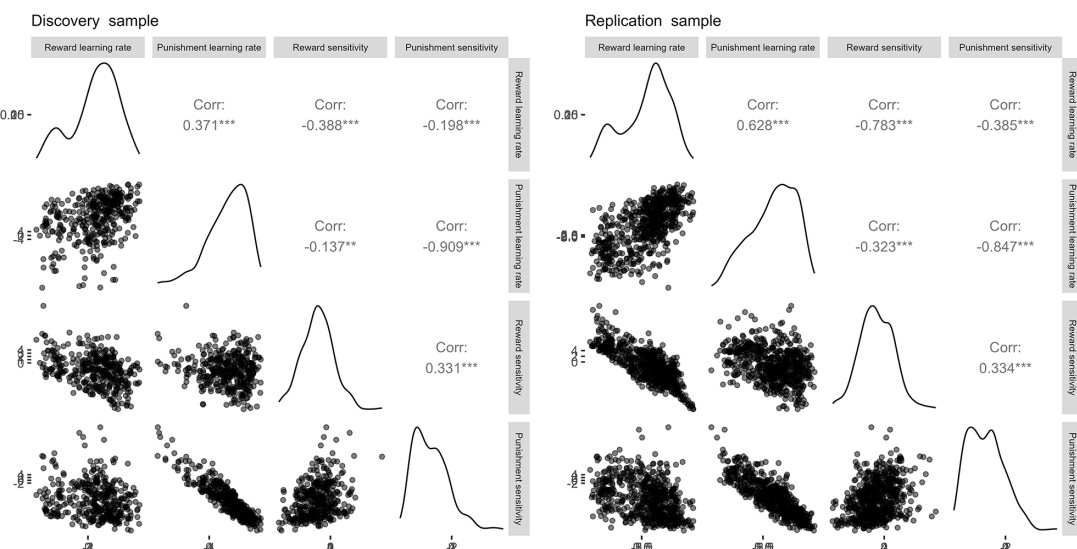

**Appendix 2—figure 2.** Correlation matrices for the estimated parameters across studies. Lower-right diagonal of each matrix shows a scatterplot of cross-parameter correlations. Upper-right diagonal denotes the Pearson's *r* correlation coefficients for each pair of parameters, based on the untransformed parameter values.

## Appendix 3

### Mediation analyses

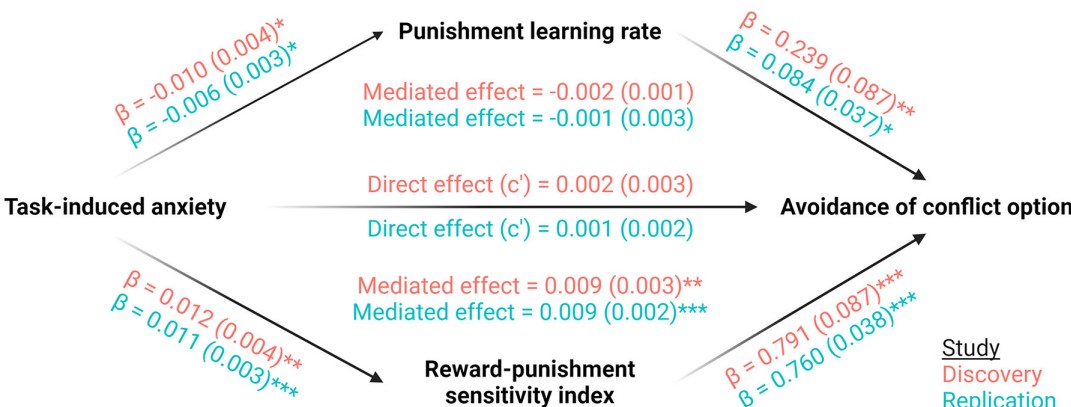

**Appendix 3—figure 1.** Mediation analyses across studies. Mediation effects were assessed using structural equation modelling. Bold terms represent variables and arrows depict regression paths in the model. The annotated values next to each arrow show the regression coefficient associated with that path, denoted as *coefficient (standard error)*. Only the reward-punishment sensitivity index significantly mediated the effect of task-induced anxiety on avoidance. Significance levels in all figures are shown according to the following: $p < 0.05$ – *; $p < 0.01$ – **; $p < 0.001$ – ***.

## Appendix 4

### Parameter recovery

To establish how well we could recover our parameters from the best fitting model, we simulated data from each study (discovery, replication) and refitted these new datasets with the identical model fitting procedure used for the empirical data. We matched the number of participants, number of trials per participant, and latent outcome probabilities in the simulated datasets to the empirical data, and used the best-fitting parameters for each participant to generate their choices. Trial-by-trial outcomes were generated stochastically according to the latent outcome probabilities, which meant that the simulated participants did not necessarily receive the same series of outcomes to our real participants. To average over this stochasticity in the outcomes, we repeated the simulations 100 times for each study. After fitting our best-fitting model to the simulated data, we computed Pearson's correlation coefficients across the data-generating parameters, and the recovered parameters. Across all parameters, the mean Pearson's *r* values between data-generating and recovered parameters ranged from 0.77 to 0.86, indicating excellent recovery (*Appendix 4—figure 1*).

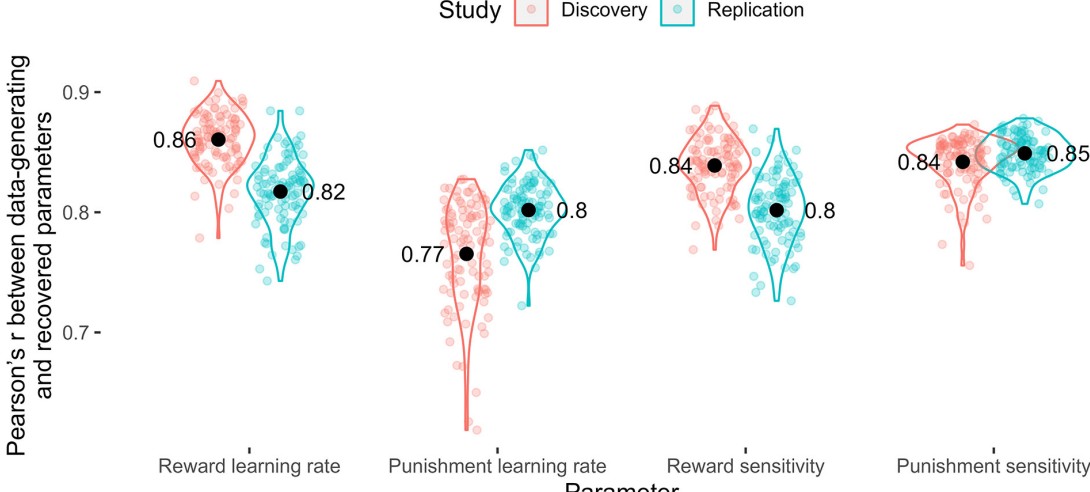

**Appendix 4—figure 1.** Parameter recovery. Pearson's *r* values across the data-generating and recovered parameters by parameter. Coloured points represent Pearson's *r* values for each of 100 simulation iterations, and black points represent the mean value across simulations.

# Appendix 5

## Split-half reliability

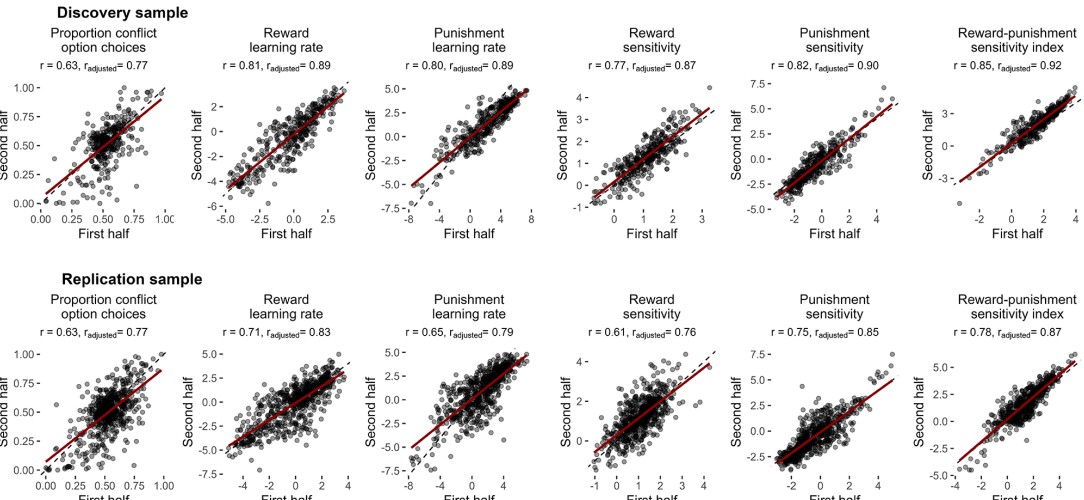

**Appendix 5—figure 1.** Split-half reliability of the task. Scatter plots of measures calculated from the first and second halves of the task are shown with their estimates of reliability (Pearson's *r* values). Reliability estimates for the computational measures from the winning computational model were computed by fitting split-half parameters within a single model, then using the parameter covariance matrix to derive Pearson's correlation coefficients for each parameter across halves. Reliability estimates are reported as unadjusted values (*r*) and after adjusting for reduced number of trials via Spearman-Brown correction ($r_{adjusted}$). Dotted lines represent the reference line, indicating perfect correlation. Red lines show lines-of-best-fit.

# Appendix 6

## Test-retest reliability

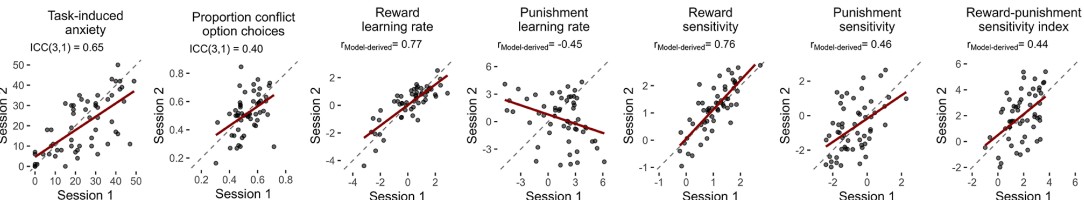

**Appendix 6—figure 1.** Test-retest reliability of the task. Scatter plots of measures calculated from the test and retest sessions are shown with their estimates of reliability (intra-class correlations: ICCs; and Pearson's *r* values). Reliability estimates for the model-agnostic measures (task-induced anxiety, proportion of conflict option choices) were estimated using intra-class correlation coefficients. Reliability estimates for the computational measures from the winning computational model were computed by first fitting both sessions' parameters within a single model, then using the parameter covariance matrix to derive a Pearson's correlation coefficient ($r_{\text{Model-derived}}$) for each parameter across sessions to be calculated from their covariance. Dotted lines represent the reference line, indicating perfect correlation. Red lines show lines-of-best-fit.

# Appendix 7

## Practice effects

We checked for practice effects in the behavioural data by comparing the behavioural measures and computational model parameters across sessions via paired t-tests. These tests were all non-significant ($p > 0.07$) except for task-induced anxiety ($t_{56} = 2.21$, $p = 0.031$) and the punishment learning rate ($t_{56} = 2.24$, $p = 0.029$; *Appendix 7—figure 1*).

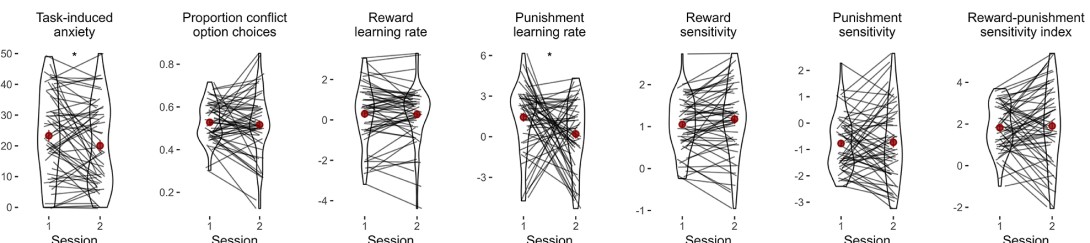

**Appendix 7—figure 1.** Task practice effects. Comparison of behavioural measures and model parameters across time. Lines represent individual data, red points represent mean values, and red lines represent standard error bars. P-values of paired t-tests are annotated above each plot. Task-induced anxiety and the punishment learning rate was significantly lower in the second session, whilst the other measures did not change significantly across sessions. Significance levels are shown according to the following: $p < 0.05$ – *.

# Appendix 8

## Sensitivity analysis: estimating parameters via EM and VBI algorithms

Given that the EM algorithm produced high inter-parameter correlations, we ran a sensitivity analysis by assessing the robustness of our computational findings to an alternative method of parameter estimation – (mean-field) VBI via Stan (**Stan Development Team, 2023**). Since, unlike EM, the results of VBI are very sensitive to initial values, we fitted the data 10 times with different initial values.

### Inter-parameter correlations

The VBI produced lower inter-parameter correlations than the EM algorithm (**Appendix 8—figure 1**).

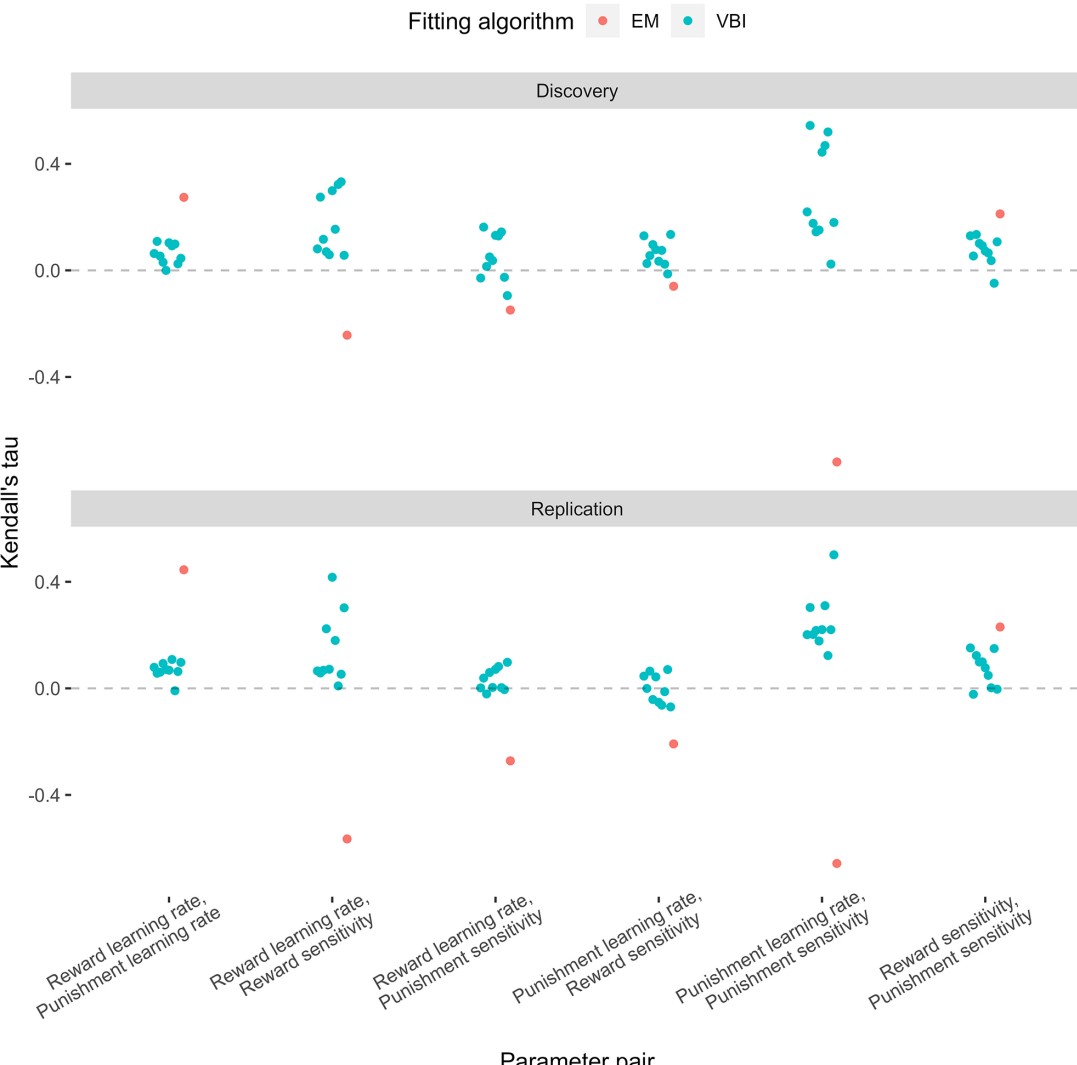

**Appendix 8—figure 1.** Inter-parameter correlations across the expectation maximisation (EM, red) and variational Bayesian inference (VBI, blue) algorithms. Overall, the VBI algorithm produced lower correlations compared to EM.

### Sensitivity analysis

Since multicollinearity in the VBI-estimated parameters was lower than for EM, indicating less trade-off in the estimation, we re-tested our computational findings from the manuscript as part of a sensitivity analysis. We first assessed whether we observed the same correlations between task-induced anxiety and punishment learning, and reward-punishment sensitivity index (**Appendix 8—figure 2a**). Punishment learning rate was not significantly associated with task-induced anxiety in any

of the 10 VBI iterations in the discovery sample, although it was in 9/10 in the replication sample. On the other hand, the reward-punishment sensitivity index was significantly associated with task-induced anxiety in 9/10 VBI iterations in the discovery sample and all iterations in the replication sample. This suggests that the correlation of anxiety and sensitivity index is robust to these two fitting approaches.

We also re-estimated the mediation models, where in the EM-estimated parameters, we found that the reward-punishment sensitivity index mediated the relationship between task-induced anxiety and task choice proportions (*Appendix 8—figure 2b*). Again, we found that the reward-punishment sensitivity index was a significant mediator in 9/10 VBI iterations in the discovery sample and all iterations in the replication sample. Punishment learning rate was also a significant mediator in 9/10 iterations in the replication sample, although it was not in the discovery sample for all iterations, and this was not observed for the EM-estimated parameters.

Overall, we found that our key results, that anxiety is associated with greater sensitivity to punishment over reward, and this mediates the relationship between anxiety and approach-avoidance behaviour, were robust across both fitting methods.

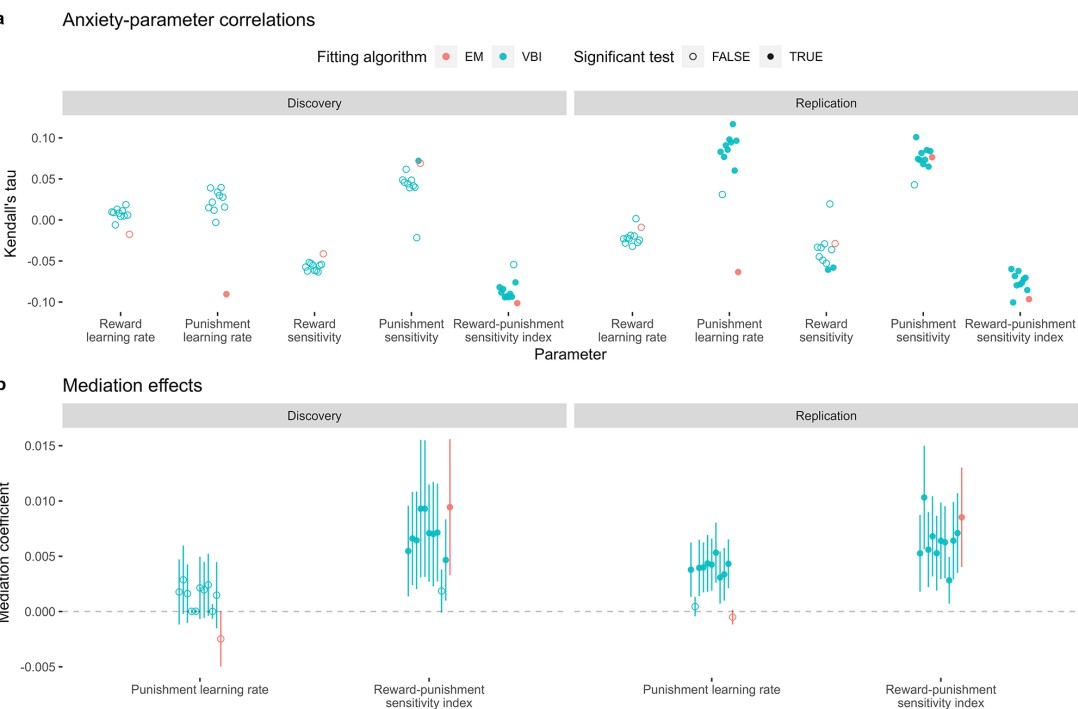

**Appendix 8—figure 2.** Sensitivity analysis of the computational findings relating to task-induced anxiety; comparing results when using parameters estimated via expectation maximisation (EM, red) and variational Bayesian inference (VBI, blue). (**a**) Kendall's tau correlations across each parameter and task-induced anxiety. (**b**) Mediating effects of the punishment learning rate and reward-punishment sensitivity index.

# Appendix 9

## Sensitivity analysis: punishment unpleasantness

### Distribution of unpleasantness

The punishments were rated as unpleasant by the participants, on average (discovery sample: mean rating = 31.1 [scored between 0 and 50], SD = 13.1; replication sample: mean rating = 32.1, SD = 12.7; *Appendix 9—figure 1*).

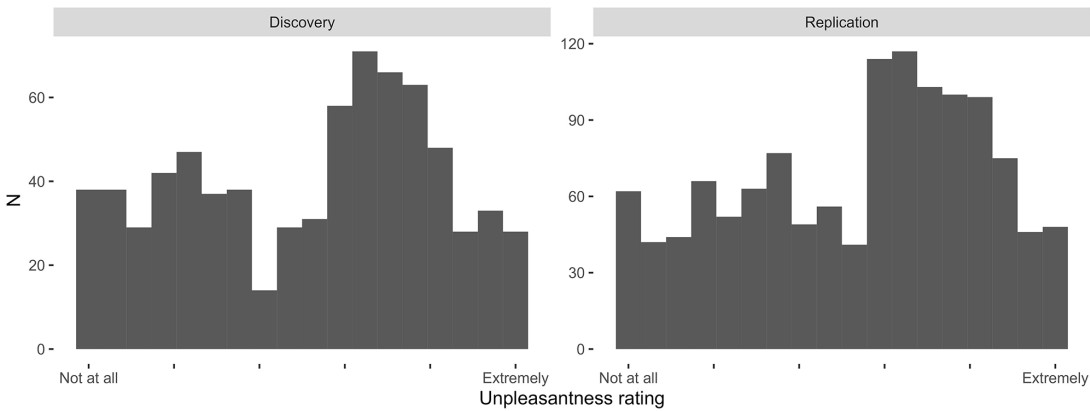

**Appendix 9—figure 1.** Distribution of self-reported punishment unpleasantness ratings. Ratings were scored from 'Not at all' to 'Extremely' (encoded as 0 and 50, respectively). Distributions are shown across the discovery and replication samples.

### Approach-avoidance hierarchical logistic regression model

We assessed whether approach and avoidance responses, and their relationships with state anxiety, were impacted by punishment unpleasantness, by including unpleasantness ratings as a covariate into the hierarchical logistic regression model. Whilst unpleasantness was a significant predictor of choice (positively predicting safe option choices), all significant predictors and interaction effects from the model without unpleasantness ratings survived (*Appendix 9—figure 2*). Critically, this suggests that punishment unpleasantness does not account for all of the variance in the relationship between anxiety and avoidance.

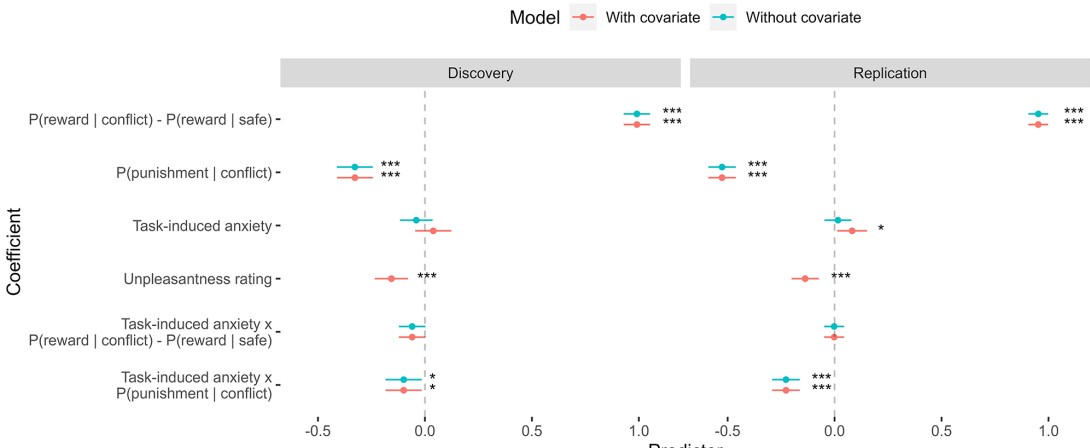

**Appendix 9—figure 2.** The effect of including unpleasantness ratings as a covariate in the hierarchical logistic regression models of task choices. Dots represent coefficient estimates from the model, with confidence intervals. Models are shown for both the discovery and replication samples. Significance levels are shown according to the following: $p < 0.05 – *; p < 0.01 – **; p < 0.001 – ***$.

## Mediation model

When unpleasantness ratings were included in the mediation models, the mediating effect of the reward-punishment sensitivity index did not survive (discovery sample: standardised $\beta$ = 0.003 ± 0.003, $p$ = 0.416; replication sample: standardised $\beta$ = 0.004 ± 0.003, $p$ = 0.100; **Appendix 9— figure 3**). Pooling the samples resulted in an effect that narrowly missed the significance threshold (standardised $\beta$ = 0.004 ± 0.002, $p$ = 0.068).

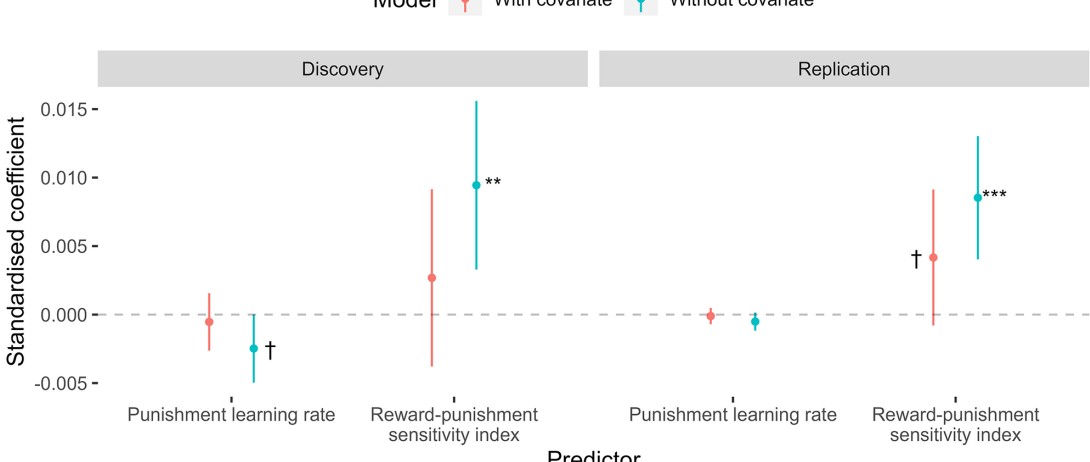

**Appendix 9—figure 3.** The effect of including unpleasantness ratings as a covariate in the mediation models. Dots represent coefficient estimates from the model, with confidence intervals. Models are shown for both the discovery and replication samples. Significance levels are shown according to the following: $p < 0.1 – †$; $p < 0.05 – *$; $p < 0.01 – **$; $p < 0.001 – ***$.

## Test-retest reliability of unpleasantness

The test-retest reliability of unpleasantness ratings was excellent (ICC(3,1) = 0.75), although participants gave significantly lower ratings in the second session ($t_{56}$ = 2.7, $p$ = 0.008, $d$ = 0.37; mean difference of 3.12, SD = 8.63).

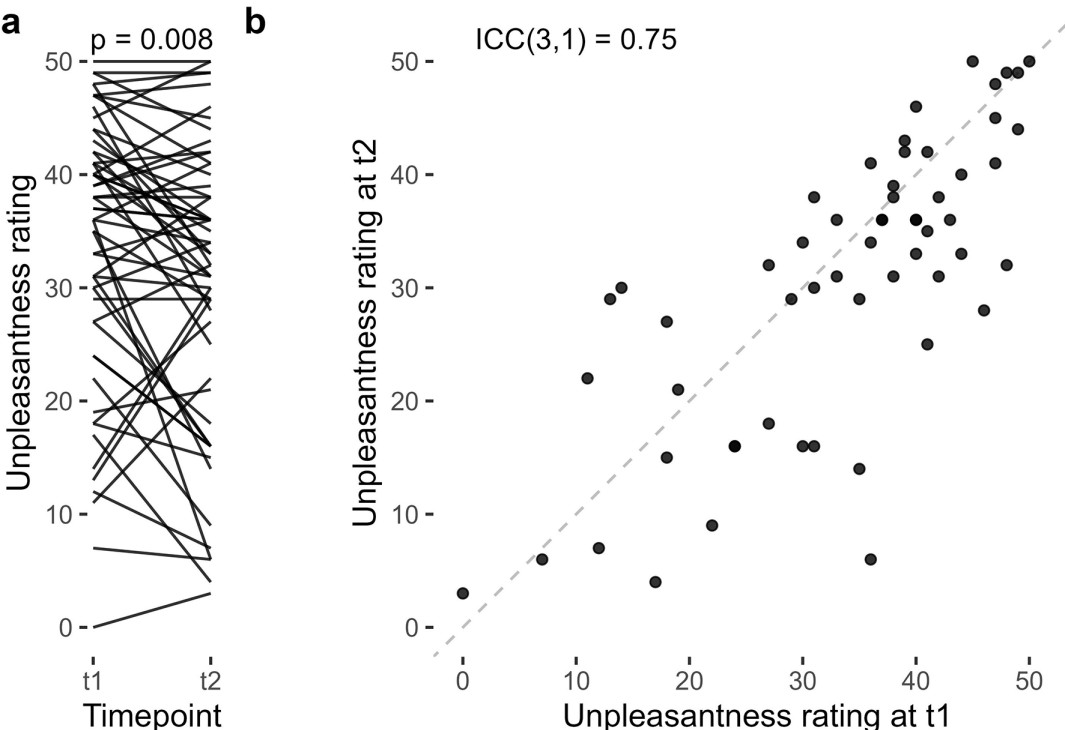

**Appendix 9—figure 4.** Test-retest reliability of unpleasantness ratings. (**a**) Comparing unpleasantness ratings across timepoints, participants rated the punishments as significantly less unpleasant in the second session. (**b**) Correlation of ratings across timepoints.

## Reliability of other measures with/out unpleasantness

To assess the effect of accounting for unpleasantness ratings on reliability estimates of task performance, we extracted variance components from linear mixed models, following a standard approach (*Nakagawa et al., 2017*) – note that this was not the method used to estimate reliability values in the main analyses, but we used this specific approach to compare the reliability values with and without the covariate of unpleasantness ratings. The results indicated that unpleasantness ratings did not have a material effect on reliability (*Appendix 9—figure 5*).

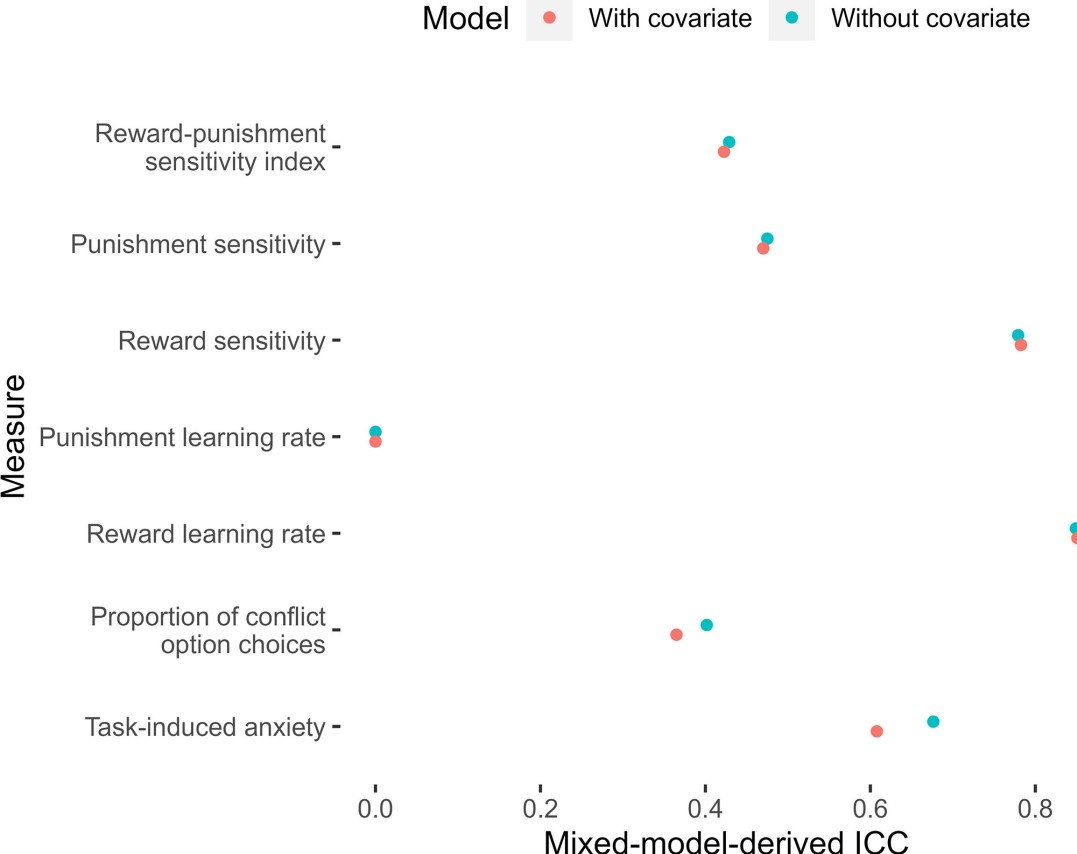

**Appendix 9—figure 5.** Mixed-model-derived intra-class correlation coefficients (ICCs) for measures of task performance, with and without accounting for unpleasantness. Dots represent model-derived ICCs.

## Appendix 10

### Sensitivity analysis: anxiety and inflexibility

As anxious participants were slower to update their estimates of punishment probability, we determined whether this was due to greater general inflexibility by examining the model including two sensitivity parameters, but one general learning rate (i.e. not split by outcome). The correlation between this general learning rate and task-induced anxiety was not significant in either samples (discovery: $\tau = -0.02$, $p = 0.504$; replication: $\tau = -0.01$, $p = 0.625$), suggesting that the effect is specific to punishment.

## Appendix 11

### Effects of outcome probabilities on choice in the task: non-linear effects

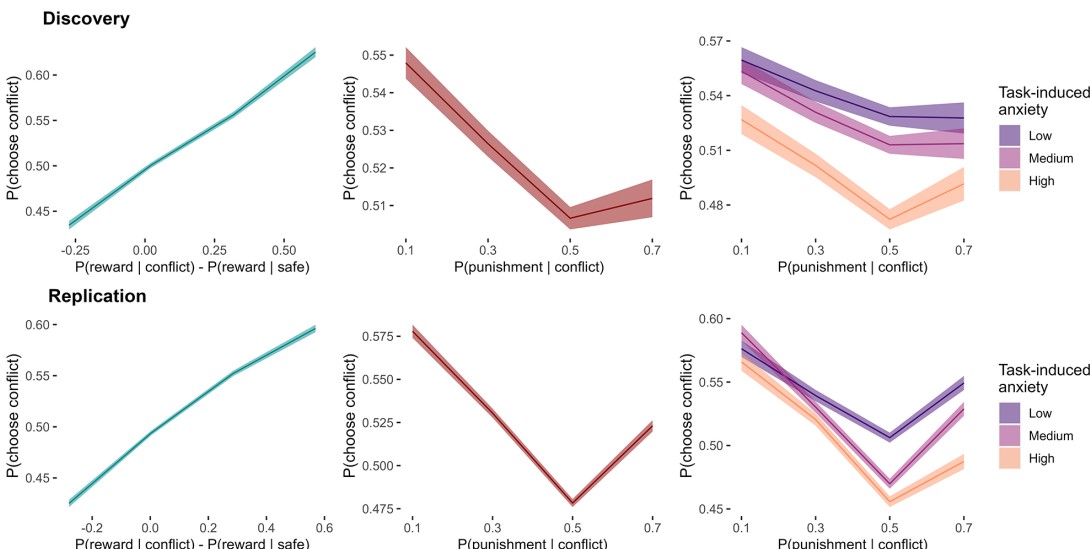

**Appendix 11—figure 1.** Effects of outcome probabilities on proportion of conflict option choices. Mean probabilities of choosing the conflict arm across the sample are plotted with standard errors. The relationships between the drifting outcome probabilities in the task and group choice proportions showed non-linear trends in both the discovery and replication samples, especially for the effect of punishment probability on choice (both main effect and interaction effect with anxiety). *Note.* For visualisation purposes, the continuous predictors (based on the latent outcome probabilities or task-induced anxiety) were categorised into discrete bins.

