## [Editor Report · eLife assessment]

This is a **valuable** paper demonstrating the validity of a novel task that could advance the field of reinforcement learning to better incorporate threat processing in approach-avoidance-conflict. A **compelling** methodology includes the use of online samples and computational modelling, psychometrics, discovery/replication and pre-registration. This work provides a foundation for future work, which is required to establish this task as relevant to psychopathology and treatment.

---

## [Referee Report · Reviewer #1 (Public Review)]

This paper describes the development and initial validation of an approach-avoidance task and its relationship to anxiety. The task is a two-armed bandit where one choice is 'safer' - has no probability of punishment, delivered as an aversive sound, but also lower probability of reward - and the other choice involves a reward-punishment conflict. The authors fit a computational model of reinforcement learning to this task and found that self-reported state anxiety during the task was related to a greater likelihood of choosing the safe stimulus when the other (conflict) stimulus had a higher likelihood of punishment. Computationally, this was represented by a smaller value for the ratio of reward to punishment sensitivity in people with higher task-induced anxiety. They replicated this finding, but not another finding that this behavior was related to a measure of psychopathology (experiential avoidance), in a second sample. They also tested test-retest reliability in a sub-sample tested twice, one week apart and found that some aspects of task behavior had acceptable levels of reliability. The introduction makes a strong appeal to back-translation and computational validity. The task design is clever and most methods are solid - it is encouraging to see attempts to validate tasks as they are developed. The lack of replicated effects with psychopathology may mean that this task is better suited to assess state anxiety, or to serve as a foundation for additional task development.

---

## [Referee Report · Reviewer #2 (Public Review)]

Summary:

The authors develop a computational approach-avoidance-conflict (AAC) task, designed to overcome the limitations of existing offer based AAC tasks. The task incorporated likelihoods of receiving rewards/ punishments that would be learned by the participants to ensure computational validity and estimated model parameters related to reward/punishment and task induced anxiety. Two independent samples of online participants were tested. In both samples participants who experienced greater task induced anxiety avoided choices associated with greater probability of punishment. Computational modelling revealed that this effect was explained by greater individual sensitivities to punishment relative to rewards.

Strengths:

Large internet-based samples, with discovery sample (n = 369), pre-registered replication sample (n = 629) and test-retest sub group (n = 57). Extensive compliance measures (e.g. audio checks) seek to improve adherence.

There is a great need for RL tasks that model threatening outcomes rather than simply loss of reward. The main model parameters show strong effects and the additional indices with task based anxiety are a useful extension. Associations were broadly replicated across samples. Fair to excellent reliability of model parameters is encouraging and badly needed for behavioral tasks of threat sensitivity.

The task seems to have lower approach bias than some other AAC tasks in the literature.

Appraisal and impact:

Overall this is a very strong paper, describing a novel task that could help move the field of RL forward to take account of threat processing more fully. The large sample size with discovery, replication and test-retest gives confidence in the findings. The task has good ecological validity and associations with task-based anxiety and clinical self-report demonstrate clinical relevance. Test-retest of the punishment learning parameter is the only real concern. Overall this task provides an exciting new probe of reward/threat that could be used in mechanistic disease models.

Additional context:

The sex differences between the samples are interesting as effects of sex are commonly found in AAC tasks. It would be interesting to look at the main model comparison with sex included as a covariate.

---

## [Referee Report · Reviewer #3 (Public Review)]

This study investigated cognitive mechanisms underlying approach-avoidance behavior using a novel reinforcement learning task and computational modelling. Participants could select a risky "conflict" option (latent, fluctuating probabilities of monetary reward and/or unpleasant sound [punishment]) or a safe option (separate, generally lower probability of reward). Overall, participant choices were skewed towards more rewarded options, but were also repelled by increasing probability of punishment. Individual patterns of behavior were well-captured by a reinforcement learning model that included parameters for reward and punishment sensitivity, and learning rates for reward and punishment. This is a nice replication of existing findings suggesting reward and punishment have opposing effects on behavior through dissociated sensitivity to reward versus punishment.

Interestingly, avoidance of the conflict option was predicted by self-reported task-induced anxiety. Importantly, when a subset of participants were retested over 1 week later, most behavioral tendencies and model parameters were recapitulated, suggesting the task may capture stable traits relevant to approach-avoidance decision-making.

The revised paper commendably adds important additional information and analyses to support these claims. The initial concern that not accounting for participant control over punisher intensity confounded interpretation of effects has been largely addressed in follow-up analyses and discussion.

This study complements and sits within a broad translational literature investigating interactions between reward/punishers and psychological processes in approach-avoidance decisions.

---

## [Author Response]

The following is the authors’ response to the previous reviews

**Reviewer #1:**
Comment on revised manuscript: Thank you for your responses - they have addressed most of my concerns.

We thank the reviewer again for their assistance in improving our manuscript.

**Reviewer #2:**
Additional context:The sex differences between the samples are interesting as effects of sex are commonly found in AAC tasks. It would be interesting to look at the main model comparison with sex included as a covariate.

Firstly, we thank the reviewer for their re-evaluation of our manuscript.

To the reviewer’s comment, we apologise for the lack of clarity. The analyses included in our revision were indeed based on the main logistic regression model of choice, including sex and age as covariates. We have clarified this in the manuscript as follows:

*While sex was significantly associated with choice in the hierarchical logistic regression in the discovery sample (β = 0.16 ± 0.07, p = 0.028) with males being more likely to choose the conflict option, this pattern was not evident in the replication sample (β = 0.08 ± 0.06, p = 0.173), and age was not associated with choice in either sample (p > 0.2).*

As it is difficult to include sex as a covariate in the reinforcement learning models in the classical sense as in a linear regression, we assessed sex effects on the individual parameters produced by these models instead, as follows:

*Comparing parameters across sexes via Welch’s t-tests revealed significant differences in reward sensitivity (t289 = -2.87, p = 0.004, d = 0.34; lower in females) and consequently reward-punishment sensitivity index (t336 = -2.03, p = 0.043, d = 0.22; lower in females i.e. more avoidance-driven). In the replication sample, we observed the same effect on reward-punishment sensitivity index (t626 = -2.79, p = 0.005, d = 0.22; lower in females). However, the sex difference in reward sensitivity did not replicate (p = 0.441), although we did observe a significant sex difference in punishment sensitivity in the replication sample (t626 = 2.26, p = 0.024, d = 0.18).*

Could the authors double check the mean/SD of approach in each group for typos? The numbers are identical.

Thank you for spotting this – the means were indeed similar (discovery: 0.521, replication: 0.516), but the standard deviations were marginally different (discovery: 0.140, replication: 0.148). We have amended the manuscript to reflect this, as follows:

*Across individuals, there was considerable variability in overall choice proportions (discovery sample: mean = 0.52, SD = 0.14, min/max = [0.03, 0.96]; replication sample: mean = 0.52, SD = 0.15, min/max = [0.01, 0.99]).*

**Reviewer #3:**
The revised paper commendably adds important additional information and analyses to support these claims. The initial concern that not accounting for participant control over punisher intensity confounded interpretation of effects has been largely addressed in follow-up analyses and discussion.I commend the authors on their revisions. My initial concerns have been largely addressed. Minor suggestions below.

We thank the reviewer again for their assistance in improving our analyses and manuscript.

Changing the visualisation of the logistic regression model in Figure 2 to tertiles instead of quartiles seems expedient, and does not properly address the points raised by the other reviewers. The argument that non-linear trends in the extreme bins are due to less data is plausible, but unsatisfying given how reliable the pattern seems to be (across samples, with small standard error) and . It is possible, albeit perplexing, that the influence of punishment probability on choice is non-linear. I think the current figure with tertiles is acceptable, but I would suggest including the figures with non-linear data as a supplementary figure, for sake of transparency and reader interest.

We agree that this is likely more complex than a simple linear effect (in the logistic space), especially given the concurrent reward probabilities which also fluctuate in the task. We also agree that the non-linear figures should be made available in the interests of transparency, and have included them in the Supplementary Materials.

We direct interested readers to the relevant section from the figure legend as follows:

*"Figure 2. Predictors of choice in the approach-avoidance reinforcement learning task. … We show linear curves here since these effects were estimated as linear effects in the logistic regression models, however the raw data showed non-linear trends – see Supplementary Figure 15."*

We have included the non-linear figures in Supplementary Section 9.11 Effects of outcome probabilities on choice in the task: non-linear effects as Supplementary Figure 15.

As an aside, the argument that approach-avoidance joystick tasks do not have a non-human counterpart misconstrues the translational root of these tasks, which was (at least in part) an attempt to model (successfully or not) general approach/avoidance processes measured in non-human tasks, e.g. appetitive/aversive runway tasks using rodents.

Our aim in this manuscript was to develop a task that was closely matched to non-human counterparts in both the experimental procedure (choice over reward/punishment outcomes) and cognitive process involved (simultaneous reward/punishment learning). With this in mind, we wanted to convey that non-human and human measures of approach/avoidance processes were historically distinct in terms of the procedures (e.g. using a joystick vs navigating a runway, due to ethological differences), and that this was potentially problematic with respect to computational validity. However, at this early point in the introduction, it was unnecessary to make a strong distinction between these tasks, which as the reviewer duly notes, follow similar approach/avoidance principles and share similar experimental roots. Therefore, we have opted to omit the reference to translational similarity in the relevant text, as follows:

*In humans, on the other hand, approach-avoidance conflict has historically been measured using questionnaires such as the Behavioural Inhibition/Activation Scale (Carver and White 1994), or cognitive tasks that rely on motor/response time biases, for example by using joysticks to approach/move towards positive stimuli and avoid/move away from negative stimuli (Guitart-Masip, Huys et al. 2012, Phaf, Mohr et al. 2014, Kirlic, Young et al. 2017, Mkrtchian, Aylward et al. 2017).*